# Recurrent network model for learning goal-directed sequences through reverse replay

**Tatsuya Haga\*, Tomoki Fukai\***

RIKEN Center for Brain Science, Wako, Japan

**Abstract** Reverse replay of hippocampal place cells occurs frequently at rewarded locations, suggesting its contribution to goal-directed path learning. Symmetric spike-timing dependent plasticity (STDP) in CA3 likely potentiates recurrent synapses for both forward (start to goal) and reverse (goal to start) replays during sequential activation of place cells. However, how reverse replay selectively strengthens forward synaptic pathway is unclear. Here, we show computationally that firing sequences bias synaptic transmissions to the opposite direction of propagation under symmetric STDP in the co-presence of short-term synaptic depression or afterdepolarization. We demonstrate that significant biases are created in biologically realistic simulation settings, and this bias enables reverse replay to enhance goal-directed spatial memory on a W-maze. Further, we show that essentially the same mechanism works in a two-dimensional open field. Our model for the first time provides the mechanistic account for the way reverse replay contributes to hippocampal sequence learning for reward-seeking spatial navigation.
DOI: https://doi.org/10.7554/eLife.34171.001

## Introduction

The hippocampus plays an important role in episodic memory and spatial processing in the brain (*O'Keefe and Dostrovsky, 1971*; *Scoville and Milner, 1957*). Because an episode is a sequence of events, sequential neural activity has been extensively studied as the basis of hippocampal memory processing. In the rodent hippocampus, firing sequences of place cells are replayed during awake immobility and sleep (*Carr et al., 2011*; *Pfeiffer, 2018*) and these reactivations are crucial for performance in spatial memory tasks (*Girardeau et al., 2009*; *Jadhav et al., 2012*; *Singer et al., 2013*). Replay can be either in the same firing order as experienced (forward replay) or in the reversed order (reverse replay). Forward replay is observed during sleep after exploration (*Lee and Wilson, 2002*; *Wikenheiser and Redish, 2013*) or in immobile states before rats start to travel towards reward (*Diba and Buzsáki, 2007*; *Pfeiffer and Foster, 2013*), hence forward replay is thought to engage in the consolidation and retrieval of spatial memory. In contrast, reverse replay presumably contributes to the optimization of goal-directed paths because rewarded spatial paths are replayed around the timing of reward delivery (*Diba and Buzsáki, 2007*; *Foster and Wilson, 2006*), and the occurrence frequency is modulated by the presence and the amount of reward (*Ambrose et al., 2016*; *Singer and Frank, 2009*).

Because sequences are essentially time asymmetric, sequence learning often hypothesizes asymmetric spike-timing-dependent plasticity (STDP) found in CA1 (*Bi and Poo , 2001*) which induce long-term potentiation (LTP) for pre-to-post firing order and long-term depression (LTD) for post-to-pre firing order. This type of STDP enables a recurrent network to reactivate sequential firing in the same order with the experience. However, in the hippocampal area CA3, it was recently reported that the default form of STDP is time symmetric at recurrent synapses (*Mishra et al., 2016*). Because CA3 is the most likely source of hippocampal firing sequences (*Middleton and McHugh, 2016*;

\*For correspondence:
tatsuya.haga@riken.jp (TH);
tfukai@riken.jp (TF)

**Competing interests:** The authors declare that no competing interests exist.

**eLife digest** To find their way around, animals – including humans – rely on an area of the brain called the hippocampus. Studies in rodents have shown that certain neurons in the hippocampus called place cells become active when an animal passes through specific locations. At each position, a different set of place cells fires. A journey from A to B will thus be accompanied by a sequential activation of place cells corresponding to a particular point.

Rats can learn new routes to a given place. Every time they take a specific way, the connections between the activated place cells become stronger. After learning, the hippocampus replays the sequence of place cell activation both in the order the rat has experienced and backwards. This is known as reverse replay, which occurs more often when the animals find rewards at their destination. This suggests that reverse replay may help animals learn the routes to locations where food is available.

To test this idea, Haga and Fukai built a computer model that simulates the hippocampal activity seen in a rat running through a maze. In contrast to previous models, which featured only forward replay, the new simulation also included reverse replay. The results confirmed that reverse replay helps the rodents to learn routes to rewarded locations. It also enables the hippocampus to combine multiple past experiences, which may teach animals that a combination of previous paths will lead to a reward, even if they have never tried the combined route before.

The hippocampus has a central role in many different types of memory. The findings by Haga and Fukai may therefore provide a framework for studying the mechanisms of memory and decision-making. The results could even offer insight into the mechanisms of logical thinking. After all, the ability to combine multiple known paths into a new route bears some similarity to joining up thought processes such as 'If Sophie oversleeps, she will miss the bus' with 'If Sophie misses the bus, she will be late for school' to reason that 'If Sophie oversleeps, she will be late for school'. Future studies should test whether reverse replay helps with this process of deduction.

DOI: https://doi.org/10.7554/eLife.34171.002

*Nakashiba et al., 2009*), this finding raises the question whether and how STDP underlies sequence learning in hippocampus. A symmetric time window implies that a firing sequence equally strengthens both forward synaptic pathways leading to the rewarded location and reverse pathways leaving away from the rewarded location in CA3 recurrent network. However, reward-based optimization requires selective reinforcement of forward pathways as it will strengthen prospective place-cell sequences in subsequent trials and forward replay events in the consolidation phase. Such bias toward forward sequences in post-experience sleep (*Wikenheiser and Redish, 2013*) and goal-directed behavior (*Johnson and Redish, 2007*; *Pfeiffer and Foster, 2013*; *Wikenheiser and Redish, 2015*) is actually observed in hippocampus. How this directionality arises in replay events and how reverse replay enables the learning of goal-directed navigation remain unclear.

In this paper, we first show how goal-directed path learning is naturally realized through reverse replay in a one-dimensional chain model. To this end, we hypothesize that the contribution of presynaptic spiking for STDP is attenuated in CA3 by short-term depression, as was revealed in the rat visual cortex (*Froemke et al., 2006*). Under this condition, symmetric STDP and a rate-based Hebbian plasticity rule bias recurrent synaptic weights toward the opposite direction to the propagation of a firing sequence, implying that the combined rule virtually acts like anti-Hebbian STDP. We also show that accumulation of afterdepolarization (*Mishra et al., 2016*) in postsynaptic neurons results in the same effect. By simulating the model with various spiking patterns, we confirm this effect for a broad range of spiking patterns and parameters of plasticity rules, including those observed in experiments. Based on this mechanism, we built a two-dimensional recurrent network model of place cells with the combination of reverse replay, Hebbian plasticity with short-term plasticity, and reward-induced enhancement of replay frequency (*Ambrose et al., 2016*; *Singer and Frank, 2009*). We first demonstrate that the network model can learn forward pathways leading to reward on a W-shaped track. Further, we extend the role of reverse replay to unbiased sequence propagations from reward sites on a two-dimensional open field, which enable learning of goal-directed behavior in the open field. Unlike the previous models for hippocampal sequence learning (*Blum and Abbott,*

*1996*; *Gerstner and Abbott, 1997*; *Jahnke et al., 2015*; *Jensen and Lisman, 1996*; *Sato and Yamaguchi, 2003*; *Tsodyks and Sejnowski, 1995*) in which recurrent networks learn and strengthen forward sequences through forward movements, our model proposes goal-directed path learning through reverse sequences.

## Results

### Hebbian plasticity with short-term depression potentiates reverse synaptic transmissions

We first simulated a sequential firing pattern that propagates through a one-dimensional recurrent neural network, and evaluated weight changes by rate-based Hebbian plasticity rules. The network consists of 500 rate neurons, which were connected with distance-dependent excitatory synaptic weights modulated by short-term synaptic plasticity (STP) (*Romani and Tsodyks, 2015*; *Wang et al., 2015*). In addition, the network had global inhibitory feedback to all neurons. A first external input to a neuron at one end (#0) elicited traveling waves of neural activity propagating to the opposite end (*Figure 1A*). Here, we regard these activity patterns as a model of hippocampal firing sequences (*Romani and Tsodyks, 2015*; *Wang et al., 2015*). A second external input to a neuron at the center of the network triggered firing sequences propagating to both directions (*Figure 1A*) because synaptic weights were symmetric (*Figure 1B*). Here, we implemented a standard Hebbian plasticity rule, which potentiated excitatory synaptic weights by the product of postsynaptic and presynaptic neural activities. During the propagation of the first unidirectional sequence, this rule potentiated synaptic weights symmetrically without creating any bias in the synaptic weights (*Figure 1B*).

However, when synaptic weights were changed by a modified Hebbian plasticity rule (see Materials and methods), in which the long-term plasticity is also regulated by STP at presynaptic terminals (*Froemke et al., 2006*), the second firing sequence only propagated to the reverse direction of the first one (*Figure 1C*). This selective propagation occurred because the first firing sequence potentiated synaptic weights asymmetrically in the forward and reverse directions, thus creating a bias in the spatial distribution of synaptic weights (*Figure 1D*). This means that firing sequences strengthen the reverse synaptic transmissions more strongly than the forward ones in this model, and reverse sequences are more likely to be generated after forward sequences.

Why does this model generate such a bias to the reverse direction? To explain this, we show the packet of neural activity during the first firing sequence (at $t = 300 \ \mathrm{ms}$) in *Figure 1E*, top. We also plotted neurotransmitter release from the presynaptic terminal of each neuron (presynaptic outputs), which is determined by the product of presynaptic neural activity and the amount of available neurotransmitters in the combined Hebbian rule (*Figure 1E*, bottom). Because neurotransmitters are exhausted in the tail of the activity packet, presynaptic outputs are effective only at the head of the activity packet. Due to this spatial asymmetricity of presynaptic outputs, connections from the head to the tail are strongly potentiated but those from the tail to the head are not (*Figure 1F*). This results in the biased potentiation of reverse synaptic transmissions in this model (*Figure 1G*). In other words, the packet of presynaptic outputs becomes somewhat 'prospective' (namely, slightly skewed toward the direction of activity propagation), so the weight changes based on the coincidences between presynaptic outputs and postsynaptic activities result in the selective potentiation of connections from the 'future' to the 'past' in the firing sequence. This mechanism was not known previously and enables forward sequences to potentiate the synaptic transmissions responsible for reverse replay, and vice versa.

In the above simulations, STP contributed crucially to the asymmetric potentiation of synaptic connections between neurons. Meanwhile, similar asymmetricity, and hence the potentiation of reverse synaptic transmission, can also emerge when the effect of postsynaptic activity on Hebbian plasticity accumulates through time (*Figure 1—figure supplement 1*). Because STDP in CA3 depends on the afterdepolarization (ADP) in dendrites (*Mishra et al., 2016*), this phenomenon will occur if the effect of ADPs accumulates over multiples postsynaptic spikes as afterhyperpolarization modulates STDP in the visual cortex (*Froemke et al., 2006*). Because we can obtain essentially the same results for STP-dependent and ADP-dependent plasticity rules, we only consider the effect of STP in the rest of this paper.

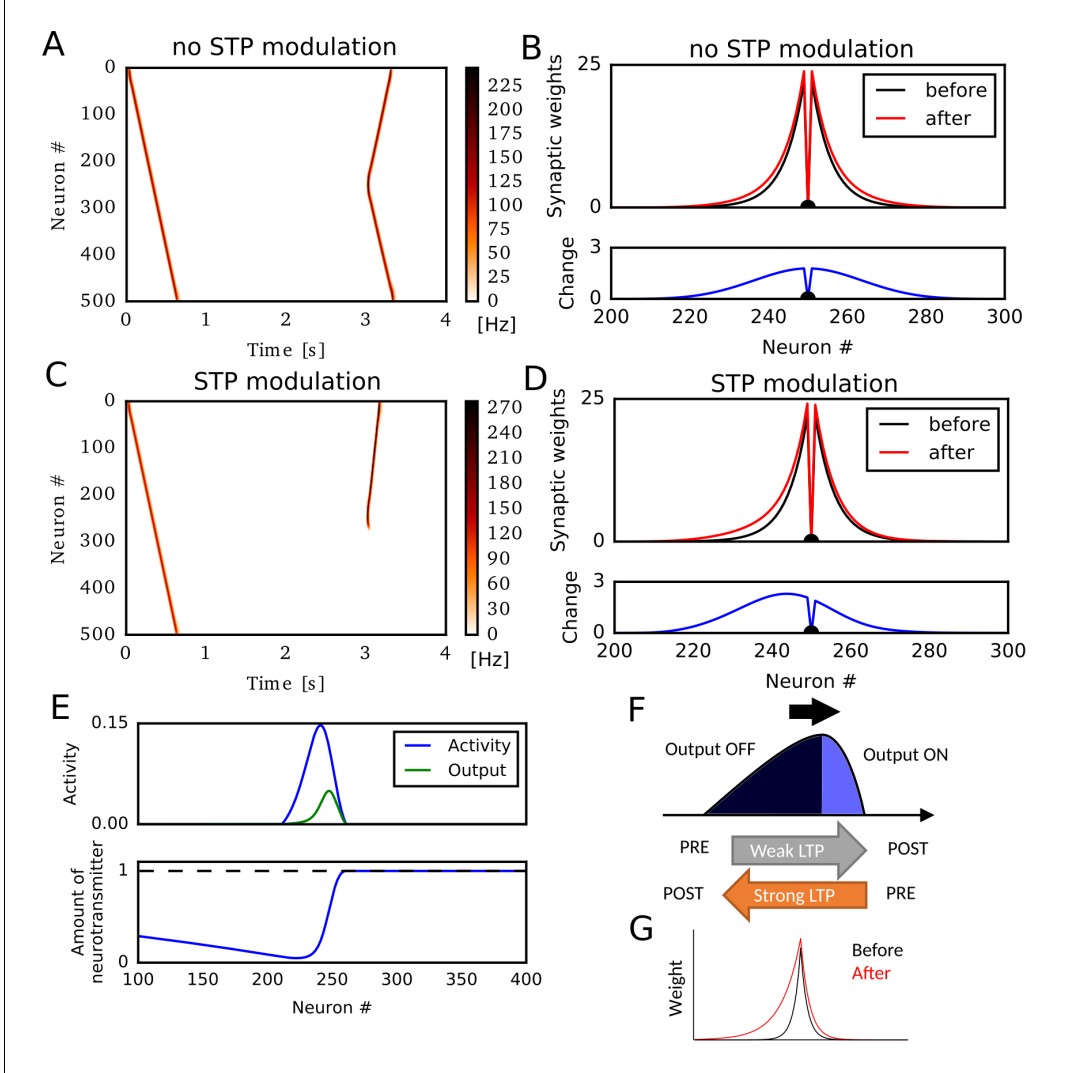

**Figure 1.** Potentiation of reverse propagation in 1-D recurrent network of rate neurons. (**A, C**) Weight modifications were induced by a firing sequence in a 1-D recurrent network (at time 0 s), and the effects of these changes on sequence propagation were examined at time 3 s. Recurrent synaptic weights were modified by the standard Hebbian plasticity rule (**A**) or the revised Hebbian plasticity rule, where the latter was modulated by STP. (**B, D**) Weights of outgoing synapses from neuron #250 to other neurons are shown at time 0 s (black) and 3 s (red) of the simulations in (**A**) and (**C**), respectively (top). Lower panels show the weight changes (blue). (**E**) Neural activity, presynaptic outputs (top) and the amount of neurotransmitters at presynaptic terminals (bottom) are shown at 300 ms of the simulation in (**C**). (**F**) Activity packet traveling on the 1-D recurrent network and the resultant weight changes by the revised Hebbian plasticity rule are schematically illustrated. (**G**) Distributions of synaptic weights are schematically shown before and after a sequence propagation.

DOI: https://doi.org/10.7554/eLife.34171.003

The following figure supplement is available for figure 1:

**Figure supplement 1.** Potentiation of reverse propagation in 1-D recurrent network with accumulation of afterdepolarization (ADP).

DOI: https://doi.org/10.7554/eLife.34171.004

## Potentiation of reverse synaptic transmissions by STDP

The potentiation of reverse synaptic transmissions also occurs robustly in spiking neurons with STDP. To show this, we constructed a one-dimensional recurrent network of Izhikevich neurons (*Izhikevich, 2003b*, *2004*) connected via conductance-based AMPA and NMDA synaptic currents (see Materials and methods). We chose Izhikevich model because its frequency adaptation induces instability and enhances the generation of moving activity bumps. However, we note that the learning mechanism itself does not depend on a specific model of spiking neurons. Initial synaptic

weights, STP, and global inhibitory feedback were similar to those used in the rate neuron model. We tested two types of STDP: asymmetric STDP in which pre-to-post firing leads to potentiation and post-to-pre firing leads to depression, and symmetric STDP in which both firing orders result in potentiation if two spikes are temporally close or depression if two spikes are temporally distant. Experimental evidence suggests that recurrent synapses in hippocampal CA3 undergo symmetric STDP (*Mishra et al., 2016*).

Among these STDP types, only symmetric STDP showed a similar effect to the rate-based Hebbian rule. Under symmetric STDP without modulation by STP, synaptic transmissions were potentiated in both directions (*Figure 2A and B*). However, symmetric STDP modulated by STP biased the weight changes to the reverse direction (*Figure 2C*), and accordingly the second firing sequence selectively traveled towards the opposite direction to the first sequence (*Figure 2D*). This effect did not depend significantly on synaptic time constants, and we could obtain the same effect when we turned off NMDA current and shortened time constants for AMPA and inhibition (*Figure 2—figure supplement 1A,B*). In contrast, under asymmetric STDP without modulation by STP, firing sequences strengthened the forward sequence propagation (*Figure 2—figure supplement 1C*) and the related synaptic connections (*Figure 2—figure supplement 1D*), as expected. Introducing modulation by STP did not change the qualitative results (*Figure 2—figure supplement 1E,F*). These results indicate that a greater potentiation of reverse synaptic transmissions in CA3 occurs under the modified symmetric STDP.

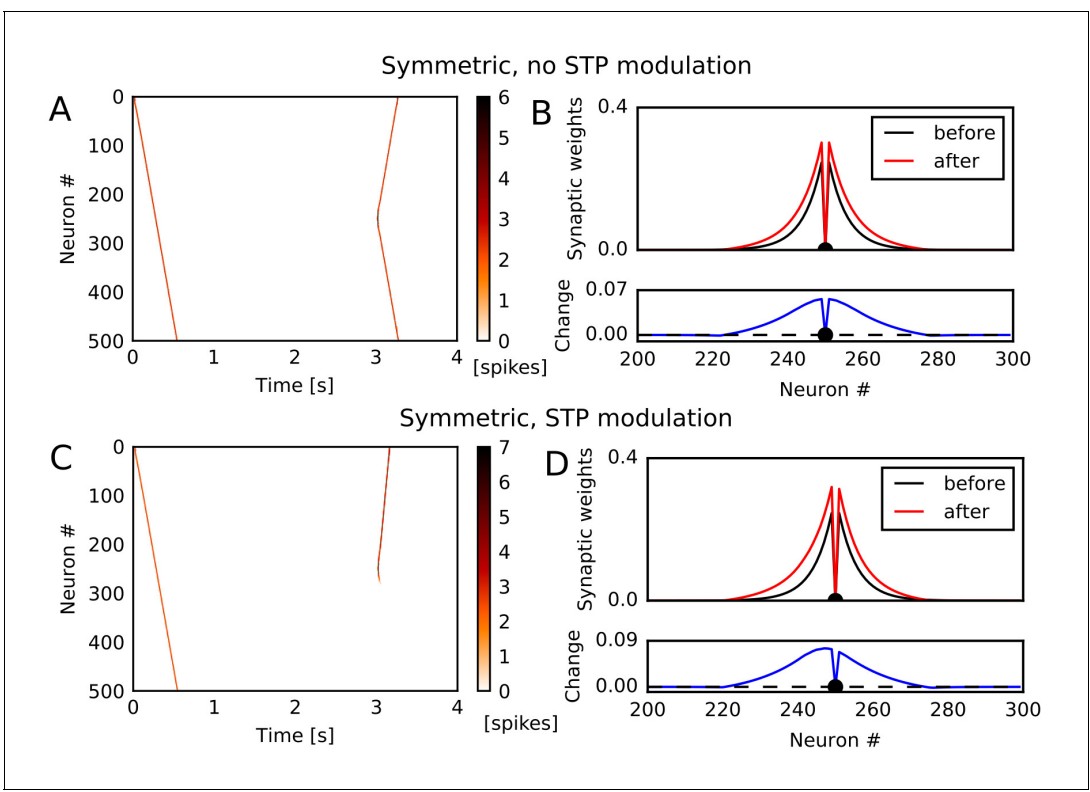

**Figure 2.** Potentiation of reverse propagation in 1-D recurrent network of spiking neurons. (**A, C**) Simulations similar to those shown in *Figure 1* were performed in a 1-D spiking neural network. Recurrent synaptic weights were changed by symmetric STDP. In (**A**) the plasticity rules were not modulated by STP whereas in (**C**) the rules were modulated. (**B, D**) The weights of outgoing synapses from neuron #250 (top) are shown at time 0 s (black) and 3 s (red) of the simulation settings shown in (**A**) and (**C**), respectively. Lower panels display the weight changes (blue).
DOI: https://doi.org/10.7554/eLife.34171.005

The following figure supplement is available for figure 2:

**Figure supplement 1.** Simulation of 1-D spiking neural network in various conditions.
DOI: https://doi.org/10.7554/eLife.34171.006

# Evaluation of parameter dependence in Poisson spike trains

We further confirmed the bias effect of the modified symmetric STDP in broader conditions than in the above network simulation. To this end, we generated sequential firing patterns along the one-dimensional network by sampling from a Poisson process, while manually controlling the number of propagating spikes per neuron, the mean inter-spike interval (ISI) of Poisson input spike trains, and time lags in spike propagation (i.e. time difference between the first spikes of neighboring neurons) (*Figure 3A*). The amount of neurotransmitter release by each presynaptic spike was calculated by the STP rule (see Materials and methods), and the magnitude of long-term synaptic changes was calculated by a Gaussian-shaped symmetric STDP (*Figure 3B*) (*Mishra et al., 2016*). The net effect of synaptic plasticity was given as the product of the two quantities, as in the previous simulations. The parameter values of short-term and long-term plasticity were adopted from experimental results

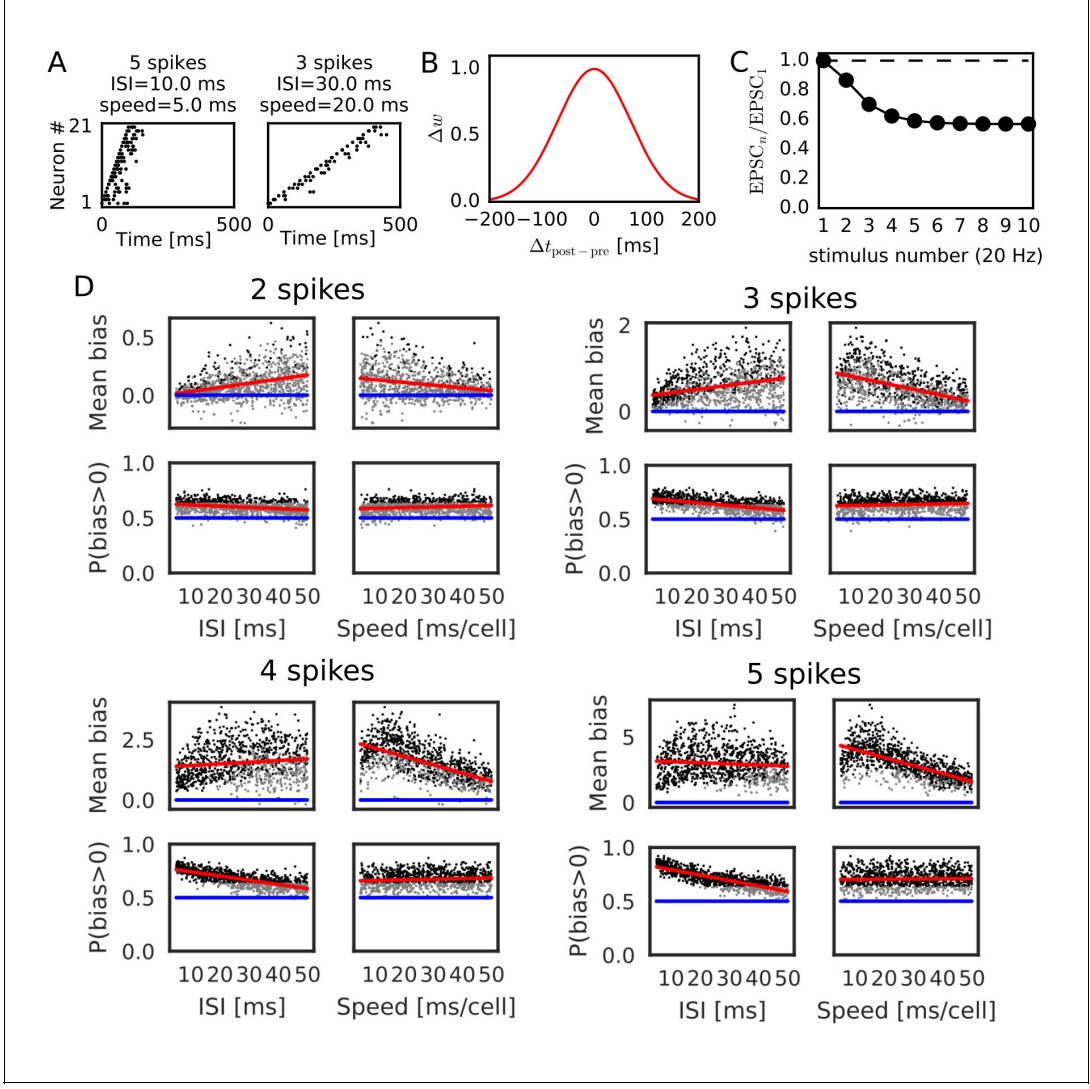

**Figure 3.** Reversely biased weight changes for symmetric STDP against variations in firing patterns. (**A**) Two examples show firing sequences having different ISIs and propagation speeds. (**B**) Gaussian-shaped symmetric STDP was used in the simulations (c.f., *Figure 1* in *Mishra et al., 2016*). (**C**) Changes in the amplitude ratio of EPSC during a 20 Hz stimulation are shown (c.f., Supplementary Figure 5 in *Guzman et al., 2016*). See the text for the parameter setting. (**D**) The mean (top) and fraction (bottom) of biases towards reverse direction are plotted for various parameter settings. A more positive bias indicates stronger weight changes in the reverse direction. Black dots correspond to parameter settings giving statistically significant positive or negative biases (p<0.01 in Wilcoxon signed rank test for the mean bias or binomial test for P(bias >0)), whereas grey dots are not significant. Red lines are linear fits to the data and blue lines indicate zero-bias (mean bias = 0 and P(bias >0)=0.5). See also *Table 1*.
DOI: https://doi.org/10.7554/eLife.34171.007

(*Figure 3B,C*) (*Guzman et al., 2016*; *Mishra et al., 2016*). We calculated the long-term weight changes in synapses sent from the central neuron in the network, and defined a weight bias as the difference in synaptic weights between forward and reverse directions (in which positive values mean bias to the reverse direction). We then obtained the mean bias and the fraction of positive biases (P(bias >0)) over 100 different realizations of spike trains generated with the same parameter values. For each number of spikes per neuron (2, 3, 4 and 5 spikes), we ran 1000 simulations using different mean ISIs and time lags randomly sampled from the interval [5, 50 ms].

Significant biases toward the reverse direction were observed in broad simulation conditions (*Figure 3D*). The biases were statistically significant (p<0.01 in Wilcoxon signed rank test for mean bias or binomial test for P(bias >0)) already in a part of conditions with two spikes per neuron, and the effect became prominent as the number of spikes increased. In general, the magnitude of synaptic changes is larger for a faster spike propagation (*Figure 3D*, *Table 1*), which is reasonable because the potentiation of symmetric STDP becomes stronger as presynaptic firing and postsynaptic firing get closer in time. On the other hand, P(bias >0) was greater for a smaller mean ISI, and it did not depend on the propagation speed (*Figure 3D*, *Table 1*). Especially, all simulations showed statistically significant biases to the reverse direction regardless of the propagation speed when the number of spikes per neuron is 4 or five and the mean ISI <20 ms. This implies that the bias toward the reverse direction is the most prominent when the neural network propagates a sequence of bursts with intraburst ISIs less than 20 ms. Such bursting is actually observed in CA3 in vivo

**Table 1.** Correlations between parameters and biases to reverse direction.
Correlations (r) between the values of parameters (mean ISI and propagation speed of spike trains, parameters of short-term plasticity) and biases in recurrent synaptic weights toward the reverse direction are listed for *Figures 3* and *4* together with the p-values.

|  | r | P |
|---|---|---|
| 2 spikes, ISI – Mean bias | 0.386 | $<10^{-10}$ |
| 2 spikes, Speed – Mean bias | −0.252 | $<10^{-10}$ |
| 2 spikes, ISI – P(bias > 0) | −0.279 | $<10^{-10}$ |
| 2 spikes, Speed– P(bias > 0) | 0.156 | $7.08 \times 10^{-7}$ |
| 3 spikes, ISI – Mean bias | 0.315 | $<10^{-10}$ |
| 3 spikes, Speed – Mean bias | −0.503 | $<10^{-10}$ |
| 3 spikes, ISI – P(bias > 0) | −0.539 | $<10^{-10}$ |
| 3 spikes, Speed– P(bias > 0) | 0.108 | 0.00066 |
| 4 spikes, ISI – Mean bias | 0.125 | $7.14 \times 10^{-5}$ |
| 4 spikes, Speed – Mean bias | −0.616 | $<10^{-10}$ |
| 4 spikes, ISI – P(bias > 0) | −0.728 | $<10^{-10}$ |
| 4 spikes, Speed– P(bias > 0) | 0.104 | 0.00104 |
| 5 spikes, $U$– Mean bias | 0.914 | $<10^{-10}$ |
| 5 spikes, $\tau_{STD}$ – Mean bias | 0.236 | $<10^{-10}$ |
| 5 spikes, $\tau_{STF}$ – Mean bias | −0.0455 | 0.151 |
| 5 spikes, $U$ – P(bias>0) | 0.869 | $<10^{-10}$ |
| 5 spikes, $\tau_{STD}$– P(bias>0) | 0.264 | $<10^{-10}$ |
| 5 spikes, $\tau_{STF}$ – P(bias>0) | −0.0416 | 0.188 |
| 5 spikes, ISI – Mean bias | −0.0896 | 0.00459 |
| 5 spikes, Speed – Mean bias | −0.658 | $<10^{-10}$ |
| 5 spikes, ISI – P(bias > 0) | −0.817 | $<10^{-10}$ |
| 5 spikes, Speed– P(bias > 0) | 0.0280 | 0.376 |

DOI: https://doi.org/10.7554/eLife.34171.008

(*Mizuseki et al., 2012*) and simulations of a CA3 recurrent network model suggest that bursting plays a crucial role in propagation of firing sequences (*Omura et al., 2015*).

Parameters that regulate STP also influence the bias toward the reverse direction. Actually, these parameters largely change in the hippocampus depending on experimental settings (*Guzman et al., 2016*). Therefore, we also performed simulations with randomly sampled values of the initial release probability of neurotransmitters ($U$) and the time constants of short-term depression and facilitation ($\tau_{STD}$ and $\tau_{STF}$). Here, the number of spikes per neuron was fixed to five, and the ISI and time lag were independently sampled from the interval [5, 20 ms] in every trial. As shown in *Figure 4* and *Table 1*, both mean bias and P(bias>0) clearly depend on the initial release probability. Prominent bias was observed for $U$>0.3, and it gradually disappeared as $U$ was decreased. The bias was weakly correlated with $\tau_{STD}$, but there was almost no correlation between the bias and $\tau_{STF}$ (*Table 1*). In sum, the bias toward the reverse direction occurred robustly for a sufficiently high intraburst firing frequency and a sufficiently high release probability of neurotransmitters.

In contrast to symmetric STDP, asymmetric STDP was not effective in potentiating reversed synaptic transmissions even if it was modulated by STP. In our simulations with parameters taken from experiments in CA1 (*Bi and Poo , 2001*), such a potentiation effect was never observed for asymmetric STDP with all-to-all spike coupling for any parameter value of STP and the number of spikes per neuron (5 or 15 spikes) (*Figure 4—figure supplement 1*). However, asymmetric STDP with nearest-neighbor spike coupling (*Izhikevich and Desai, 2003a*), in which only the nearest postsynaptic spikes before and after a presynaptic spike were taken into account, generated statistically significant biases toward the reverse direction in some parameter region (*Figure 4—figure supplement 1*). In this case, large biases required large values of $U$ and $\tau_{STD}$, and a large number of spikes per neuron (15 spikes). Thus, the condition that asymmetric STDP generates the biases to the reverse direction is severely limited, although we cannot exclude this possibility.

## Bias effects induced by spike trains during run

We also tested whether the realistic activity pattern of place cells during run can induce the directional bias. When a rat passes through place fields of CA3 place cells, they typically shows bell-shape activity patterns duration of which is about 1 s and mean peak firing rate is about 13 Hz (*Mizuseki and Buzsáki, 2013*; *Mizuseki et al., 2012*). Firing of place cells is phase-locked to theta oscillation and the firing phase gradually advances as a rat moves through place fields (theta-phase

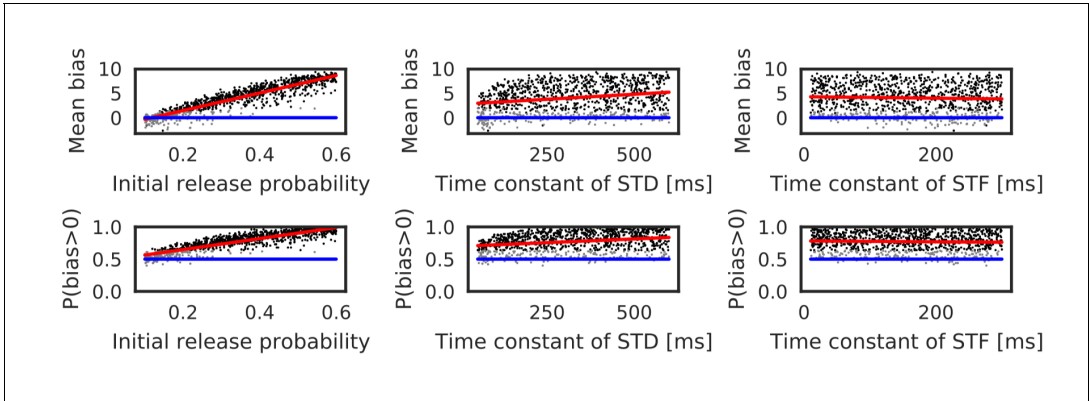

**Figure 4.** Reversely biased weight changes against variations in short-term plasticity. The mean and fraction of positive biases toward reverse direction are shown in various parameter settings of short-term plasticity, where a more positive bias indicates stronger weight changes in the reverse direction. Black dots correspond to parameter settings giving statistically significant positive or negative biases (p<0.01 in Wilcoxon signed rank test for the mean bias or binomial test for P(bias >0)), whereas grey dots are not significant. Red lines are linear fits to the data and blue lines indicate zero-bias (mean bias = 0 and P(bias >0)=0.5). See also *Table 1*.

DOI: https://doi.org/10.7554/eLife.34171.009

The following figure supplement is available for figure 4:

**Figure supplement 1.** Evaluation of biases toward reverse direction for asymmetric STDP.

DOI: https://doi.org/10.7554/eLife.34171.010

precession). Furthermore, firing sequences of place cells scan the path from behind to ahead of the rat in every theta cycle, which is a phenomenon called theta sequence (*Dragoi and Buzsáki, 2006*; *Foster and Wilson, 2007*; *Huxter et al., 2008*; *O'Keefe and Recce, 1993*; *Wang et al., 2015*; *Wikenheiser and Redish, 2015*). Some models of hippocampal sequence learning hypothesized that these compressed sequential activity patterns enhance memory formation through STDP (*Jensen and Lisman, 1996*, *2005*; *Sato and Yamaguchi, 2003*).

Here, we simulated weight biases induced by Poisson spike trains that mimic place-cell activities when a rat is running through 81 equidistantly-spaced place fields. We modified both mean peak firing rates and magnitude of theta phase-locking (phase selectivity), as shown in *Figure 5A* (see Materials and methods). We found that weight biases primarily depended on coarse-grained firing rates, but phase selectivity had no significant effect in our model (*Figure 5B*, *Table 2*). Noticeable effects

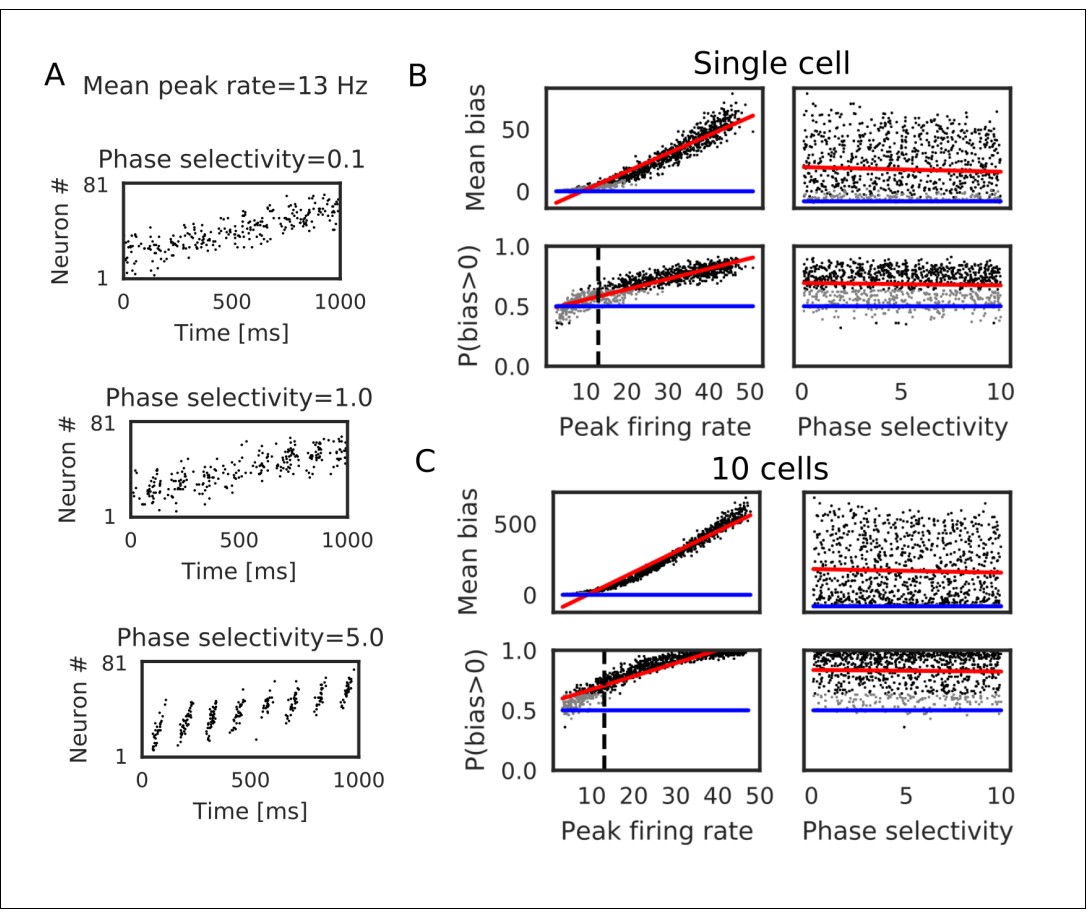

**Figure 5.** Bias effects induced by place-cell activity during run. Simulations similar to those in *Figure 3* were performed with spike trains mimicking theta sequences of place-cell firing during run. (**A**) Examples of spike trains in simulations of place-cell activity during run. We had two tuning parameters: mean peak firing rate and phase selectivity. The latter controls theta-phase locking of spikes. (**B**) The mean and fraction of positive biases to the reverse direction calculated from a single central neuron are shown in various parameter settings. A more positive bias indicates stronger weight changes in the reverse direction. Black dots correspond to the parameter settings yielding statistically significant positive or negative biases (p<0.01 in Wilcoxon signed rank test for the mean bias or binomial test for P(bias >0)), whereas grey dots are not significant. Red lines are linear fits to the data and blue lines indicate zero-bias (mean bias = 0 and P(bias >0)=0.5). (**C**) Same as B, but biases were summed over 10 neurons at the center. In B and C, the width of the STDP time window was 70 ms. See also *Table 2*.
DOI: https://doi.org/10.7554/eLife.34171.011

The following figure supplement is available for figure 5:

**Figure supplement 1.** Bias effects during run induced by narrowed symmetric STDP.
DOI: https://doi.org/10.7554/eLife.34171.012

**Table 2.** Correlations between parameters and biases during run.

Correlations (r) between the values of parameters (mean peak firing rat and phase selectivity) and biases in recurrent synaptic weights toward the reverse direction are listed for *Figure 5* and *Figure 5—figure supplement 1* together with the p-values.

| Single-cell, Broad STDP | r | P |
|---|---|---|
| Peak firing rate – Mean bias | 0.958 | $<10^{-10}$ |
| Phase selectivity – Mean bias | −0.0506 | 0.11 |
| Peak firing rate – P(bias > 0) | 0.911 | $<10^{-10}$ |
| Phase selecitivy – P(bias > 0) | −0.0478 | 0.131 |
| 10 cells, Broad STDP | r | P |
| Peak firing rate – Mean bias | 0.979 | $<10^{-10}$ |
| Phase selectivity – Mean bias | −0.0337 | 0.287 |
| Peak firing rate – P(bias > 0) | 0.926 | $<10^{-10}$ |
| Phase selectivity – P(bias > 0) | −0.0241 | 0.447 |
| Single-cell, Narrow STDP | r | P |
| Peak firing rate – Mean bias | 0.909 | $<10^{-10}$ |
| Phase selectivity – Mean bias | 0.189 | $1.85 \times 10^{-9}$ |
| Peak firing rate – P(bias > 0) | 0.939 | $<10^{-10}$ |
| Phase selectivity – P(bias > 0) | 0.0982 | 0.00186 |

DOI: https://doi.org/10.7554/eLife.34171.013

of phase selectivity on weight biases emerged for a narrow STDP time window of 10 ms (*Figure 5—figure supplement 1*, *Table 2*), suggesting that experimentally observed broad time window of STDP (70 ms) masks small differences in firing phases. As for the mean peak firing rate, 13 Hz was not high enough to induce statistically significant bias in these simulations. However, in two scenarios, weight biases can become strong during movement in our model. First, the summation of synaptic inputs from ten place cells with identical (or overlapped) place fields amplified weight biases, and the bias became significant at 13 Hz in this case (*Figure 5C*, *Table 2*). Second, firing rates of place cells obey a log-normal distribution and a small fraction of place cells exhibits extremely high firing rates (>30 Hz) (*Mizuseki and Buzsáki, 2013*). These cells created large weight changes and significant directional biases (*Figure 5B and C*). We note that the bias effects of replay sequences can be also enhanced in the above scenarios. Taken together, the bias effect in our model is weaker during run than in replay events because mean firing rate is lower and theta phase-locking is not effective for learning with broad symmetric STDP. However, non-negligible bias effects may arise in some biologically plausible situations.

## Learning strongly biased forward synaptic pathways through reverse replay

By the mechanism described above, our network model can potentiate forward synaptic pathway through reverse replay. However, whether reverse replay creates strong bias to the forward direction depends crucially on two parameters, that is, the slow time constant of long-term plasticity and the strength of short-term depression. Here, we demonstrate this by simulating the one-dimensional recurrent network model similar to *Figure 1* in three different conditions. In all the cases (*Figure 6A, B,C*), we repeatedly triggered firing sequences at the beginning of each simulation trial (0 s < time < 5 s), which are regarded as 'forward' sequences corresponding to repeated sequential experiences. After this (time >10 s), we repeatedly stimulated central neurons in the network to induce firing sequences, which selectively traveled along the reverse synaptic pathway strengthened by the forward sequences, as was demonstrated in *Figure 1* of the manuscript. These sequences overwrote the weight bias induced by forward sequences and eventually reversed it into the forward direction on neuron #100 (*Figure 6D*). In the end, the bias converges to some value for which reverse replay

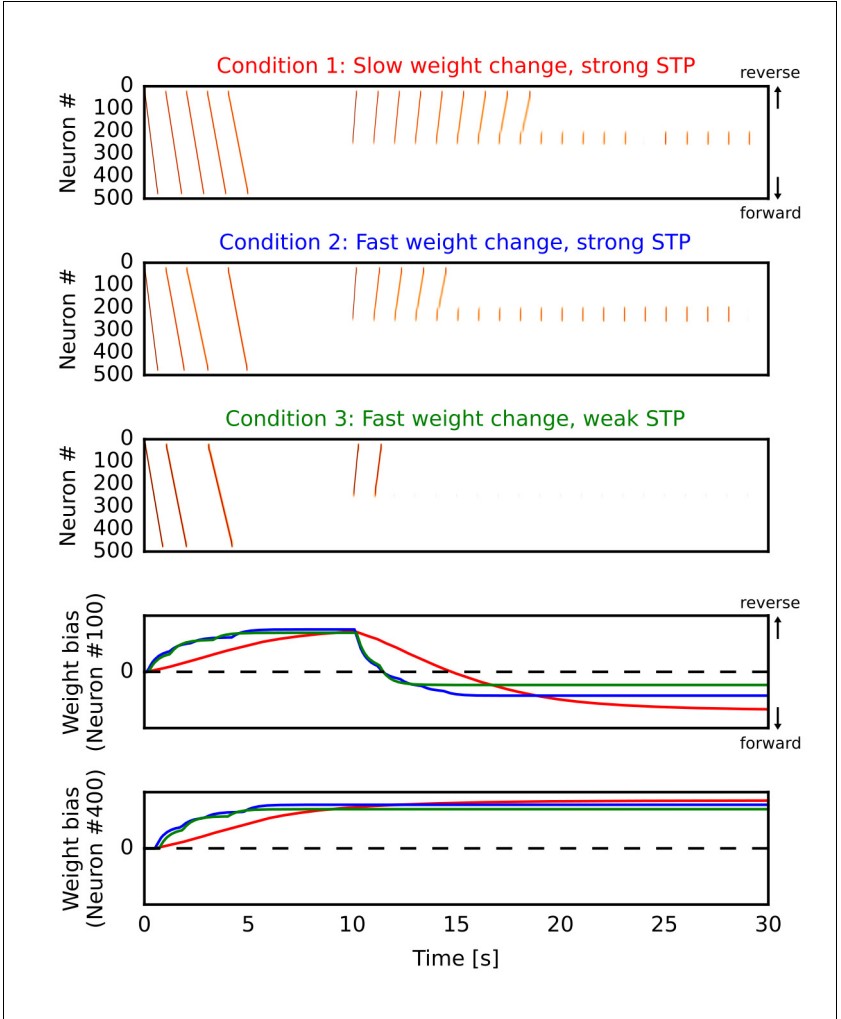

**Figure 6.** Learning a forward synaptic pathway through reverse replay. We induced forward firing sequences in the 1-D recurrent network from 0 s to 5 s, and then stimulated neurons at the center from 10 s to 30 s. We simulated three conditions in which parameters for long-term and short-term plasticity were different (see the main text). (A, B, C) Simulated neural activities in conditions 1, 2 and 3, respectively. We used a long time constant for long-term plasticity and strong short-term depression in the condition 1. Compared with the condition 1, the time constant was shortened in the condition 2 and short-term depression was also weakened in the condition 3. (D, E) Changes in the directional biases of synaptic weights on neurons #100 and #400, respectively. Red, blue and green lines correspond to the conditions 1, 2 and 3, respectively. A more positive bias indicates stronger weights in the reverse direction of the initial forward sequences.

DOI: https://doi.org/10.7554/eLife.34171.014

could not propagate a long distance. On the other hand, neuron #400 was not recruited in reverse replay and hence the reversal of weight bias did not occur (*Figure 6E*).

In the condition 1 (*Figure 6A*), modifications of synaptic weights were relatively slow (time constant was 5000 ms) and the depression effect of STP was the same as in *Figure 1*. Due to the slow modifications of synaptic weights, a large number of reverse replay was generated before the bias was overwritten. Therefore, accumulated weight bias to the forward direction became the largest in the final state (*Figure 6D*, red). In the condition 2 (*Figure 6B*), the time constant for weight changes was shorter (500 ms) and accordingly each reverse replay rapidly changed the weight bias. Consequently, reverse replay stopped earlier and the weight bias became smaller than the condition 1 (*Figure 6D*, blue). However, the bias still converges to the forward direction because firing sequences can propagate even when synaptic weights were weakly biased to the opposite direction. In the

condition 3 (*Figure 6C*), we weakened STP in addition to the short time constant for weight changes (see Materials and methods). Because short-term depression enhances the generation of firing sequences (*Romani and Tsodyks, 2015*), this manipulation further reduced the number of reverse replay and consequently the final value of weight bias (*Figure 6D*, green).

These results demonstrate that enhanced firing propagation by STP and relatively slow long-term plasticity are necessary to create strongly biased forward synaptic pathways through reverse replay. Strong short-term depression may be replaced by other mechanisms such as dendritic spikes, which also enhance the propagation of firing sequences (*Jahnke et al., 2015*). However, this possibility was not pursued in the present study.

## Goal-directed path learning through reverse replay

We now demonstrate how reverse replay events starting from a rewarded position enables the learning of goal-directed paths. We consider the case where an animal is exploring on a W-maze (*Figure 7A*). During navigation, the animal gets a reward at the one end of the arm (position D2), but not at the opposite end (position D1) and other locations. In each trial, the animal starts at the center arm (position A) and runs into one of the two side arms at position B. In the present simulations, the animal visits both ends alternately: it reaches to D1 in $(2n + 1)$-th trials and D2 in $(2n)$-th trials, where $n$ is an integer. After reaching either of the ends (i.e. D1 or D2), the animal stops there for 7 s.

We constructed the $50 \times 50$ two-dimensional (2-D) place-cell network associated to the 2-D space that the animal explored (*Figure 7B*), using the rate neuron model. Each place cell had a place field in the corresponding position on the 2-D space and received global inhibitory feedback proportional to the overall network activity. Neighboring place cells were reciprocally connected with excitatory synapses, which were modulated by short-term and long-term plasticity rules as in *Figure 1* and *Figure 6*. During the delivery of reward, we mimicked dopaminergic modulations by enhancing the inputs to CA3 and increasing the frequency of triggering firing sequences (*Ambrose et al., 2016*; *Singer and Frank, 2009*). Under this condition, a larger number of reverse replay was generated in the rewarded position. Thus, larger potentiation of synaptic pathways toward reward is expected in our model.

After a few traversals on the W-maze, the network generated reverse replay at D1 and D2 (*Figure 7D and E*, red arrows) and forward replay at A (*Figure 7D and E*, black arrows) during immobility, and theta oscillation induced theta sequences along the animal's path (see *Figure 7— video 1*). Notably, in trial five and later trials, forward replay selectively traveled towards the rewarded position D2. Furthermore, firing sequences that started from the non-rewarded position D1 propagated to the rewarded position D2 but not to the start A (*Figure 7D and E*, blue arrows). We note that the animal never traveled directly from D2 to D1 in our simulation. Thus, the network model could combine multiple spatial paths to form a synaptic pathway that has not been traversed by the animal. All these properties of firing sequences look convenient for the goal-directed learning of spatial map.

We statistically confirmed the above-mentioned biases in firing sequences. We performed independent simulations of 10 model rats, in which five rats visited the two arms in the above-mentioned order, and the other five rats visited the arms in the reversed sequential order. In each simulation, we counted the number of firing sequences propagating along different synaptic pathways. Propagation of firing sequences triggered at the start A was significantly biased to the rewarded position D2 (*Figure 7F*, Wilcoxon's signed rank test, p=$5.86 \times 10^{-3}$). While firing sequences from D2 tended to be reverse replay which propagates to A (*Figure 7G*, Wilcoxon's signed rank test, p=$5.86 \times 10^{-3}$), the majority of firing sequences from D1 propagated prospectively to D2 and hence were goal-directed sequences (*Figure 7H*, Wilcoxon's signed rank test, p=$5.57 \times 10^{-3}$).

To visualize how the recurrent network was optimized for the goal-directed exploratory behavior, we defined 'connection vectors' from recurrent synaptic weights. For each place cell, we calculated the weighted sum of eight unit vectors each directed towards one of the eight neighboring neurons, using the corresponding synaptic weights (*Figure 8A*). These connection vectors represent the average direction of neural activity transmitted from each neuron, and the 2-D vector field shows the flow of neural activities in the 2-D recurrent network and hence in the 2-D maze. We note that these vectors bias the flow, but actual firing sequences can sometimes travel in different directions from

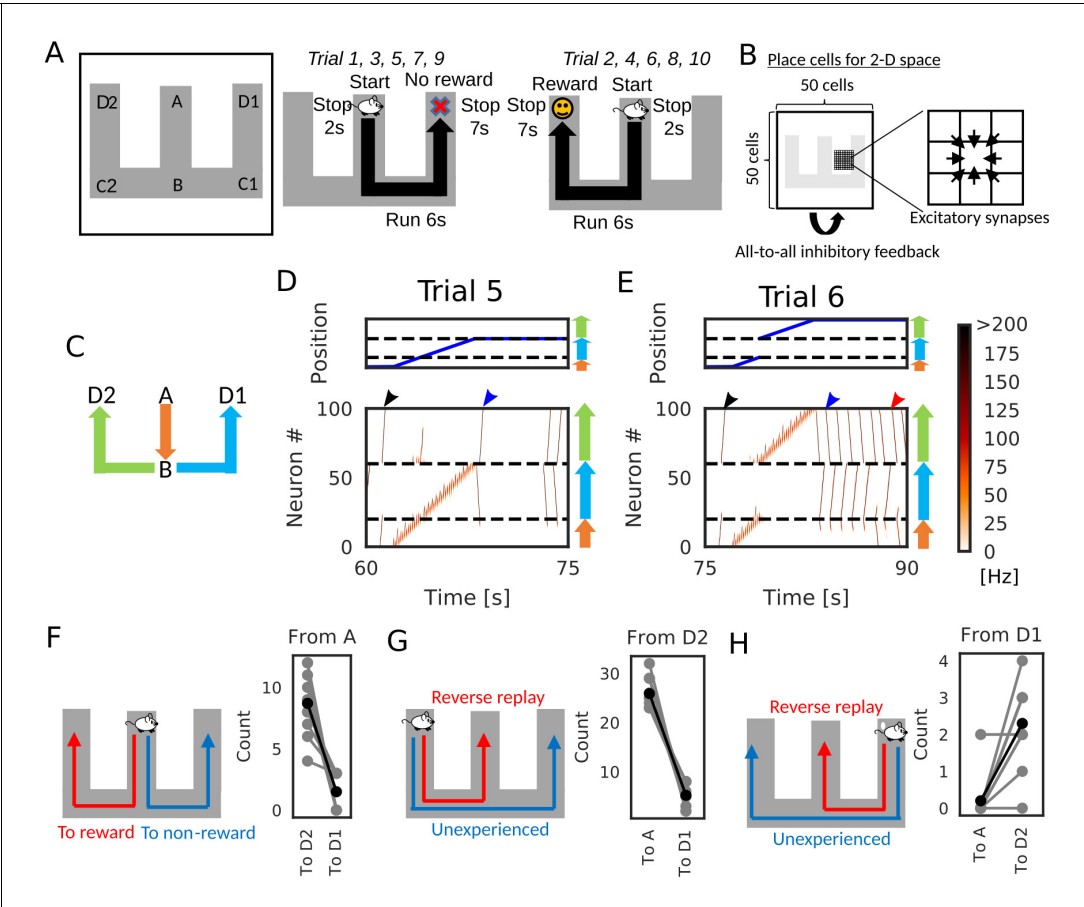

**Figure 7.** Goal-directed path learning with reverse replay on a W-maze. (**A**) An animal alternately repeats traverses from starting position A to two goal positions D1 and D2 on a track. (**B**) Schematic illustration of a 2-D recurrent network of place cells. The conjunction of receptive fields of all place cells covers the entire track. Adjacent place cells were interconnected via excitatory synapses. (**C**) A color code specifies the three portions of the W-maze in the following linearized plots. (**D, E**) Linearized plots of the animal's position (top) and simulated neural activities of place cells on the track (bottom). Black arrows indicate the end points of goal-directed sequences from A to D2 through B, a red arrow indicates the starting points of reverse replay from D2 to A through B, and blue arrows indicate the end point and starting point of replays through unexperienced paths D1 to D2 (trial 5) and D2 to D1 (trial 6), respectively. (**F, G, H**) The number of firing sequences triggered from A (**F**), from D2 (**G**) and from D1 (**H**) in each of 10 independent simulation sets (gray points). Black points show the means taken over the simulation sets.

DOI: https://doi.org/10.7554/eLife.34171.015

The following video and figure supplement are available for figure 7:

**Figure supplement 1.** Simulated place-cell activity without theta oscillation on W-maze.

DOI: https://doi.org/10.7554/eLife.34171.016

**Figure 7-Video 1.** Simulation of goal-directed path learning with reverse replay in a 2-D recurrent neural network.

DOI: https://doi.org/10.7554/eLife.34171.017

the vector flow. Initially, synaptic connections were random and the connection vector field was not spatially organized (*Figure 8—figure supplement 1*). However, after the exploration, the vector field was organized so as to route neural activities to those neurons encoding the rewarded position on the track (*Figure 8B*). A similar route map was also obtained when we reversed the sequential order of visits to the two arms (*Figure 8—figure supplement 2*) but was abolished when we removed reward (*Figure 8—figure supplement 3*) or the effect of STP on Hebbian plasticity (*Figure 8—figure supplement 4*). As demonstrated previously, direct synaptic pathways from D1 to D2 were also created. The emergence of direct paths relies on two mechanisms in this model. First, as seen in *Figure 8—figure supplement 3*, connections are biased from goal to start when there is no reward at the goal because theta sequences enhance synaptic pathways opposite to the direction of animal's movement. In non-rewarded travels, these directional biases are not overwritten by reverse

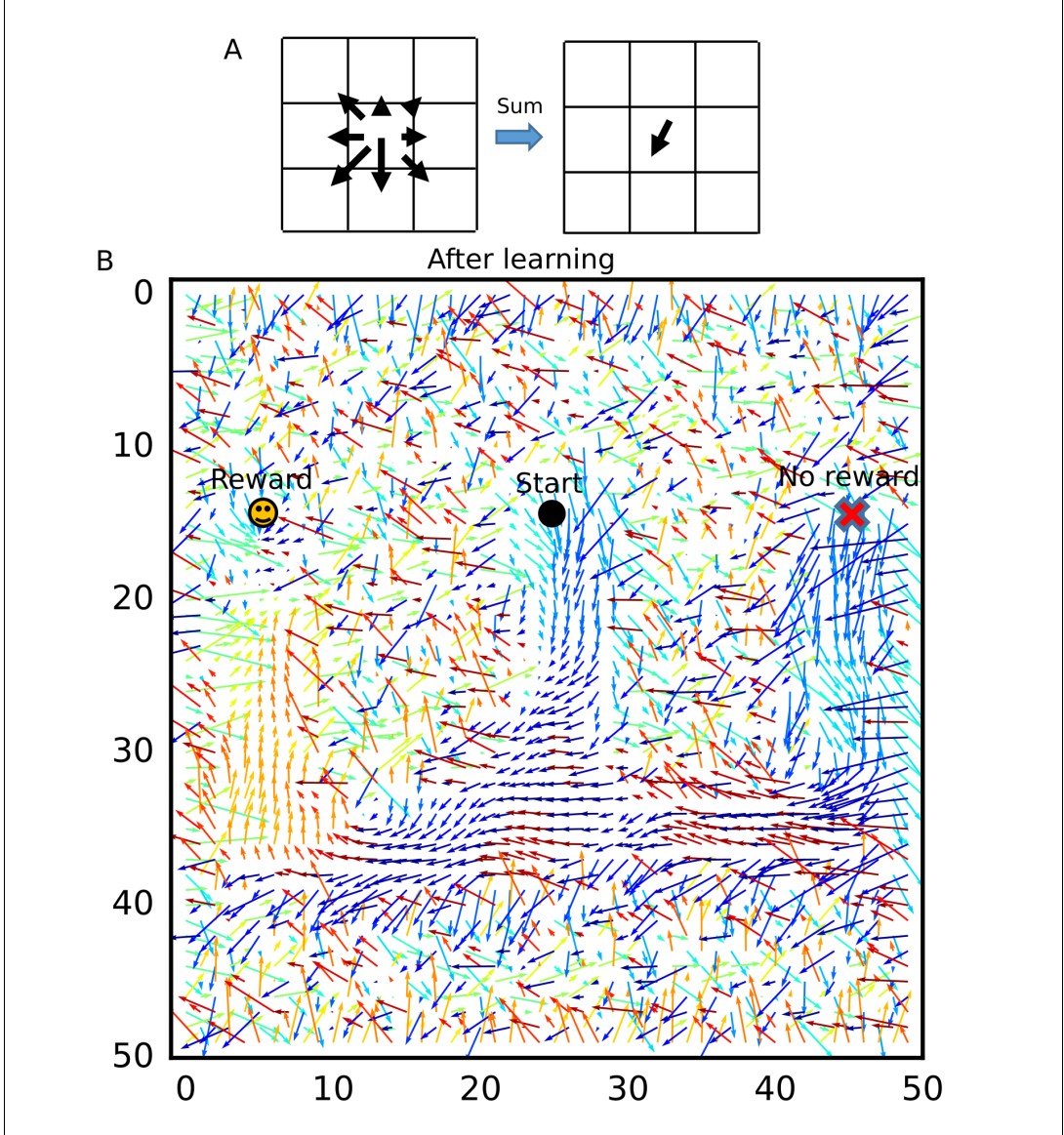

**Figure 8.** Recurrent connections organized on the W-maze for activity propagation toward reward. (**A**) Calculation of connection vectors at each position on the W-maze is schematically illustrated. Vector length in the left figure corresponds to synaptic weights. (**B**) The connection vectors formed through reverse replay point the directions leading to the rewarded goal along the track. Note that these vectors are meaningful only inside of the maze because the outside positions were not visited by the animal and the corresponding vectors merely represent the untrained initial states.
DOI: https://doi.org/10.7554/eLife.34171.018

The following figure supplements are available for figure 8:

**Figure supplement 1.** The connection vector field before exploration on the W-maze.
DOI: https://doi.org/10.7554/eLife.34171.019

**Figure supplement 2.** The connection vector field after exploration in a different order.
DOI: https://doi.org/10.7554/eLife.34171.020

**Figure supplement 3.** The connection vector field after learning without reward.
DOI: https://doi.org/10.7554/eLife.34171.021

**Figure supplement 4.** The connection vector field after learning without modulation by short-term plasticity.
DOI: https://doi.org/10.7554/eLife.34171.022

replay. Thus, the relative preference of a synaptic pathway in hippocampal sequential firing decreases for exploration that does not result in reward. Second, some of reverse replay sequences from D2 propagates into D1 instead of the stem arm (*Figure 7E and G* and *Figure 7—video 1*), and such joint replay enhances biases towards goal through unexperienced spatial paths. Thus, our model creates a map not only for the spatial paths experienced by the animal, but also for their possible combinations if they guide the animal directly to the rewarded position from a point in the space. In this sense, our model optimizes the cognitive map of the spatial environment.

We also examined the role of theta oscillation in our network model. When we turned off theta oscillation, the network model generated replay-like long-lasting firing sequences not only during immobility, but also during run (*Figure 7—figure supplement 1*). These sequences propagated randomly in both forward and reverse directions. In our model, theta oscillation offers periodic hyperpolarization of the membrane potentials that terminates firing sequences in a short period and localizes place-cell activity. Although the absence of theta oscillation does not impair place-cell sequences during run if we weaken recurrent connection weights (e.g. the model in *Wang et al., 2015*), such model does not show replay events. At least in our model, theta oscillation during run is useful to realize the robust generation of local place-cell activity during run and replay events simultaneously. We further explore the role of theta sequences in the next section.

## Unbiased sequence propagation enhances goal-directed behavior in a 2-D space

So far, we have shown the role of reverse replay for goal-directed learning in a 1-D environment. However, whether a similar mechanism works in a 2-D environment remains unclear. Previous models produce reverse replay of recent paths by transient upregulation of the excitability of recently activated neurons (*Foster and Wilson, 2006*; *Molter et al., 2006*), which may also work in a 2-D space. Such an effect has been experimentally observed, but the bias to recent paths disappears rapidly (*Csicsvari et al., 2007*). In an open arena, prospective place-cell sequences tend to propagate from the current position to reward sites, but sequence propagation from the goal position was not biased to recent paths (*Pfeiffer, 2018*; *Pfeiffer and Foster, 2013*). These observations suggest that firing sequences during immobility in a 2-D space propagate isotropically from trigger points, rather than reverse replay of recent paths.

Our model predicts that such firing sequences triggered at reward sites are beneficial for goal-directed path learning in a 2-D space. To show this, we simulated the same 2-D neural network model as in the previous section, increasing initial connection weights. Because we connected only neighboring neurons, these strong neuronal wiring reflected the topological structure of 2-D square space. We intermittently stimulated the central place cells to trigger firing sequences, which homogenously propagated through the 2-D neural network (*Figure 9A*). Because sequence propagation potentiates synaptic pathways in the opposite direction, these divergent firing sequences created the connection vector field converging to the center (*Figure 9B*). This result generalizes the role of 1-D reverse replay to higher dimensional spaces: isotropic sequence propagation from reward sites achieves goal-directed sequence learning in a 2-D (or even 3-D) open field.

We further demonstrate how this learning mechanism works for goal-directed navigation in an open arena in a task similar to Morris water maze task (*Foster et al., 2000*; *Morris et al., 1986*; *Vorhees and Williams, 2006*) and a 2-D foraging task (*Pfeiffer and Foster, 2013*). In the simulations, an animal started from random positions in a 2-D square space to search for a reward placed at one of the four candidate reward sites (*Figure 10A*). In each trial, the animal stayed at the starting position for 3 s, ran around the 2-D space at a constant speed until it found the reward, and stayed at the reward site for 15 s. We triggered replay sequences every 1 s during immobility and theta oscillation induced theta sequences during run. At each time, we calculated a vector from the animal's current position to the gravity center of the neural activity (corresponding to the current position expressed by the neural network), which we call the activity vector (*Figure 10B*). We used this vector to rotate the angle of the velocity vector of the animal's movement. The velocity vector was also updated during immobility to determine the direction of the animal's movement at the next start. Consequently, goal-directed replay sequences or theta sequences bias the animal's movement toward the goal. Such a relationship between the animal's movement and hippocampal firing sequences has been suggested in several experiments (*Huxter et al., 2008*; *Pfeiffer and Foster, 2013*; *Wikenheiser and Redish, 2015*). One simulation set consisted of 20 trials, and the position of

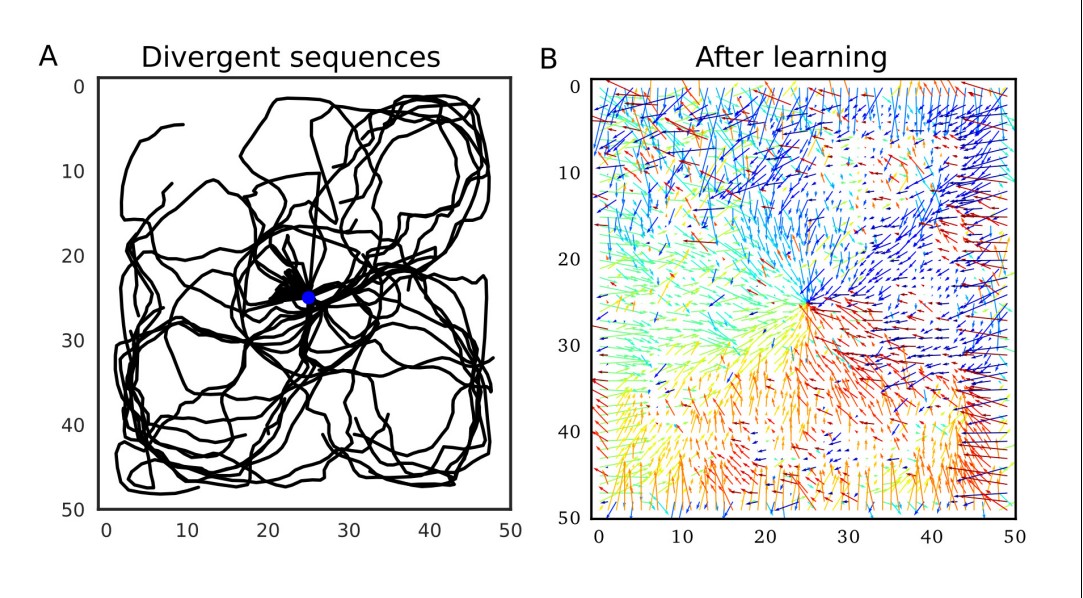

**Figure 9.** Divergent firing sequences create convergent weight biases to the triggering point. (**A**) Trajectories of divergent firing sequences triggered at the center (blue circle) in a 2-D neural network. We plotted positions of the gravity center of instantaneous neural activities during sequence propagation. (**B**) A convergent connection vector field was formed after the simulation in (**A**).
DOI: https://doi.org/10.7554/eLife.34171.023

reward was changed every five trials. Therefore, memory of the previous trials could guide the animal to the reward in trials 2–5, 7–10, 12–15, and 17–20 (REPEAT trials), but not in trials 6, 11, 16 (SWITCH trials). We performed 10 independent simulation sets, and 10 control simulation sets in which learning was disabled and the animal's behavior was similar to a random search.

The model quickly learned efficient goal-directed navigation after every SWITCH trial (*Figure 10C*). Animals took relatively short paths from start to goal in REPEAT trials (*Figure 10E*), and accordingly exploration time was significantly shorter in REPEAT trials than in other trials (*Figure 10D*, Wilcoxon's rank sum test, p<$10^{-10}$ for both REPEAT-SWITCH and REPEAT-CONTROL). In contrast, exploration time in SWITCH trials was longer than that in control simulations (*Figure 10D*, Wilcoxon's rank sum test, p=$3.43 \times 10^{-6}$) because animals typically explored around the previous reward site in SWITCH trials (*Figure 10E*). The exploration time around the previous reward site in SWITCH trials was significantly longer in simulations with learning than in control simulations (Wilcoxon's rank sum test, p=$3.21 \times 10^{-8}$), which is consistent with rodents' behavior in Morris water maze task (*Vorhees and Williams, 2006*).

To analyze biases in sequence propagation, we calculated the angle between the instantaneous activity vector and a reference vector. The reference vector was a vector from the animal's current position to a goal (reward) before and during exploration, or a vector from the animal's current position to the recent path (the animal's average position within 3 s before reaching the goal) at a goal. Bias to the goal or recent paths is strong if the angle is small. However, because of the small size of the network and the 2-D space, the angles were not exactly uniform even in control simulations. To remove this effect, we calculated a mean angular displacement in each behavioral state (start, run and goal) in control simulations and subtracted these baseline values. In REPEAT trials, the bias to reward before and during exploration was stronger than the bias to recent path at the goal (*Figure 10F*, paired sample t-test, p=$2.98 \times 10^{-7}$ for Start-Goal, p=$5.02 \times 10^{-3}$ for Run-Goal). The bias at the goal was almost zero, that is, the same level as a random search. These results suggest that the bias from start to goal was stronger than that from goal to recent paths (reverse replay) in REPEAT trials, which is consistent with experimental observation (*Pfeiffer, 2018*; *Pfeiffer and Foster, 2013*). Furthermore, the bias during exploration suggests that weight biases also affected propagation of theta sequences. In contrast, in SWITCH trials, the bias to recent paths at the goal was significantly stronger than the bias to the goal in other periods (*Figure 10G*, paired sample t-test,

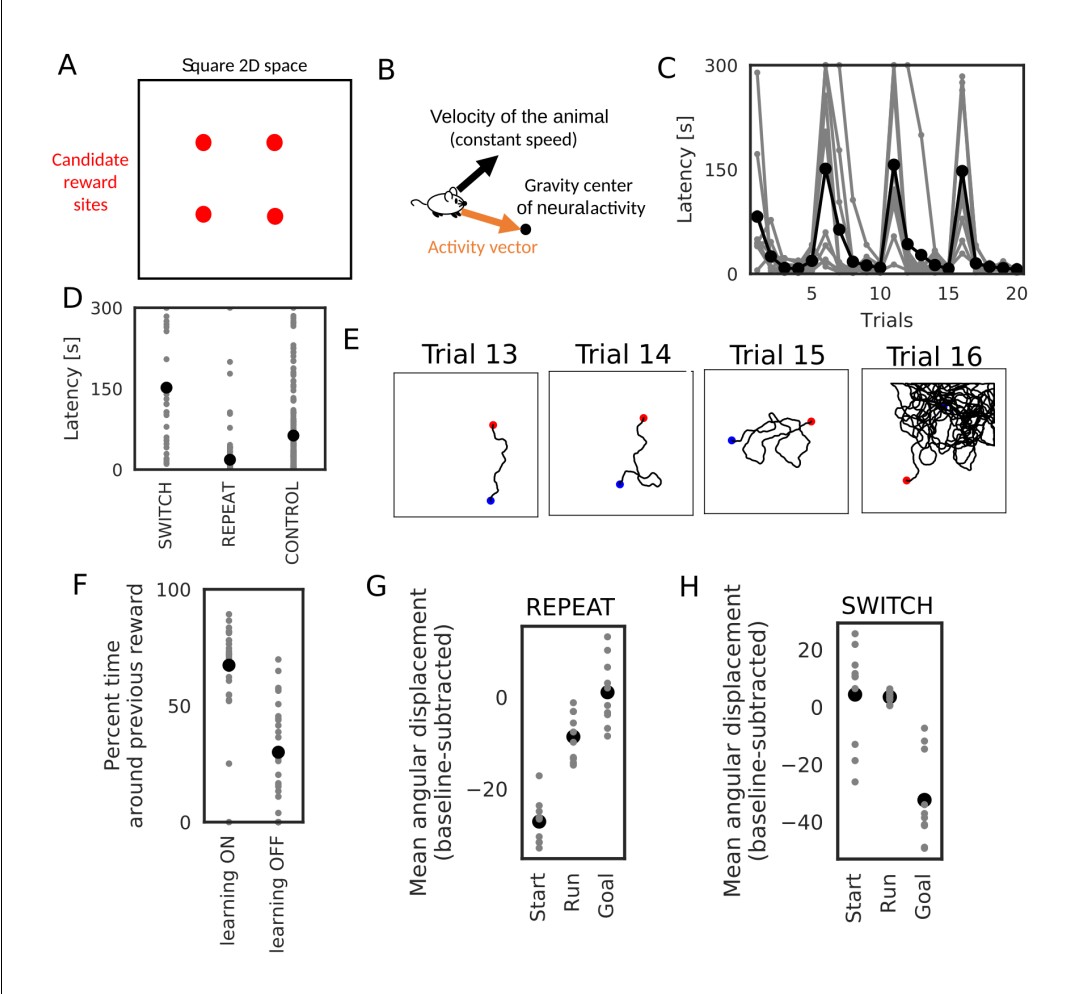

**Figure 10.** Learning goal-directed behavior in a 2-D space. In all figures, gray color shows data from individual trials or simulation sets, and black color shows means over simulations. (**A**) The square two-dimensional space in the simulation. (**B**) Computational scheme for the animal's motion. We modulated the fixed-length velocity vector with the activity vector from the animal's current position to the center of mass of neural activity in the network. (**C**) Latency to reach the goal (N = 10 simulations sets). (**D**) Comparison of latency in SWITCH trials (trial 6, 11, 16), REPEAT trials (trial 2–5, 7–10, 12–15, 16–20), and control simulations in which we turned off learning (N = 160 trials for REPEAT, N = 30 trials for SWITCH, N = 200 trials for CONTROL). (**E**) Example trajectories of the animal. Blue and red circle shows start and goal positions, respectively. (**F**) Comparison of the percent time spent in the quadrant containing the previous reward site between SWITCH trials with and without learning (N = 30 trials for both). (**G, H**) Comparison of mean angular displacements between the activity vector and reference vectors at start, during run, and goal (N = 10 simulation sets). Reference vectors were directed to reward (start and run), or directed to the recent paths (goal). We subtracted the mean angular displacements calculated from control simulations in each behavioral state.

DOI: https://doi.org/10.7554/eLife.34171.024

p=$5.9 \times 10^{-4}$ for Start-Goal in SWITCH, p=$4.83 \times 10^{-5}$ for Run-Goal in SWITCH). This bias to recent paths in SWITCH trials can be explained by the fact that the animal typically reached reward after visiting the previous reward site which strongly attracted firing sequences (*Figure 10E and F*). Notably, this bias gives an efficient way to update connection weights for the new reward site because the connection vector fields converging to the two goals differ only in the space between the goals and updating weight biases is unnecessary outside this space. Thus, our model predicts that the bias of sequence propagation from a novel goal position to previously learned goals appears when the animal should update previous memories to a new memory. Taken together, these results suggest that divergent sequences create weight biases for convergent sequence propagation to goals in our model and this learning mechanism can be a basis for efficient goal-directed navigation in the 2-D space.

## Discussion

In this paper, we showed that the modified Hebbian plasticity rule modulated by STP biases synaptic weights toward the reverse direction of the firing sequences that traveled through a recurrent network. We demonstrated this counterintuitive phenomenon in network models of rate neurons and those of spiking neurons obeying symmetric STDP. We further clarified for various Poisson-like sequential firing patterns that the phenomenon favors spike bursts and a high presynaptic release probability. We also showed that the selective potentiation of reverse directions is unlikely to occur for the conventional asymmetric STDP.

### Reverse replay and STP-modulated symmetric STDP for goal-directed learning

Our results have several implications for spatial memory processing by the hippocampus. Suppose that the animal is rewarded at a spatial position after exploring a particular path. Reverse replay propagating backward from the rewarded location will strengthen the neuronal wiring in CA3 that preferentially propagates forward firing sequences to this location along the path. Because the frequency of reverse replay increases at rewarded positions (*Ambrose et al., 2016*; *Singer and Frank, 2009*), reward delivery induces the preferential potentiation of forward synaptic pathways, which in turn results in an enhanced occurrence of forward replay in the consolidation phase. Thus, our model predicts that reverse replay is crucial for the reinforcement of reward-seeking behavior in the animal and gives, for the first time, the mechanistic account for the way reverse replay enables the hippocampal prospective coding of reward-seeking navigation. Furthermore, if the occurrence of reverse replay is modulated not only by reward but also by other salient events for the animal, this model is immediately generalized to memorization of important paths to be replayed afterwards. An interesting example would be learning of spatial paths associated with fear memory (*Wu et al., 2017*). Our computational results are qualitatively consistent with the experimentally observed properties of forward and reverse replay events (*Ambrose et al., 2016*; *Carr et al., 2011*; *Diba and Buzsáki, 2007*; *Foster and Wilson, 2006*; *Pfeiffer, 2018*; *Pfeiffer and Foster, 2013*; *Singer and Frank, 2009*) and matches the recent finding of symmetric STDP time windows in CA3 (*Mishra et al., 2016*).

Importantly, our model reinforces unexperienced spatial paths by connecting the multiple paths that were previously encoded by separate experiences. In the simulations on the W-maze, the network model not only learned actually traversed paths from the start (A) to the goal (D2), but also remembered paths from other locations (C1 and D1) to the goal despite that the animal had not experienced these paths. This reinforcement occurs because reverse replay sequences starting from the visited arm occasionally propagate or bifurcate into an unvisited arm at the branching point. The hippocampus can generate replay along joint paths (*Wu and Foster, 2014*) even when the animal has no direct experience (*Gupta et al., 2010*). Therefore, the above mechanism is biologically possible.

### Testable predictions of the model

In 1-D tracks, reverse replay is observed immediately after the first lap (*Foster and Wilson, 2006*; *Wu and Foster, 2014*). Our model shows that such a replay creates a bias to the forward direction (i.e. toward the reward) even in the very early stage of learning. Consistently, a weak bias to forward replay was observed in the first exposure to a long 1-D track (*Davidson et al., 2009*). Our model predicts that the bias to the goal-directed sequences will be suppressed (or enhanced) if we selectively block (or enhance) reverse replay at the goal. Such experiments are possible by using the techniques of real-time decoding feedback (*Ciliberti and Kloosterman, 2017*; *Sodkomkham et al., 2016*).

The most critical assumption in our model is the rapid modulation of STDP coherent to the presynaptic neurotransmitter release of STP. Such a modulation was actually reported in the visual cortex (*Froemke et al., 2006*). Although short-term depression also exists in CA3 (*Guzman et al., 2016*), STDP is modulated in a slightly different fashion in the hippocampus: the strong modulation arises from the second presynaptic spike rather than the first one (*Wang et al., 2005*). However, the experiment was performed in a dissociated culture in which hippocampal sub-regions were not distinguished. Moreover, the modulation of STDP was not tested for more than two presynaptic spikes, and whether a third presynaptic spike further facilitates or rather depresses STDP remains unknown.

Therefore, the contributions of STP to STDP should be further validated in CA3. The proposed role of short-term depression in biasing replay events may also be examined by pharmacological blockade or enhancement of STP in the hippocampus (*Froemke et al., 2006*). In addition, we showed that the dendritic ADP accumulated over multiple spikes causes a similar phenomenon (*Figure 1—figure supplement 1*). This can be directly tested in CA3 by modifying the protocol described previously (*Mishra et al., 2016*). Moreover, a bias to the reverse direction can be further strengthened if some neuromodulator expands the time window of symmetric STDP selectively toward the anticausal temporal domain ($t_{pre} > t_{post}$). Our model also predicts this type of metaplasticity in CA3

Neuromodulations, especially reward-triggered facilitation of replay events (*Ambrose et al., 2016*; *Singer and Frank, 2009*), play important roles for the proposed mechanism of goal-directed learning with reverse replay. CA3 primarily receives dopaminergic input from the locus coeruleus (LC) which signals novelty and facilitate learning in the hippocampus (*Takeuchi et al., 2016*; *Wagatsuma et al., 2018*; *Walling et al., 2012*). Therefore, sequence learning in CA3 is also affected by novelty and salience of events (*Lisman and Grace, 2005*; *Lisman et al., 2011*). Consequently, any place in which the animal experiences behaviorally important events can be a potential goal for the hippocampal path learning, and valence of the goal is encoded in downstream areas (*de Lavilléon et al., 2015*; *Redondo et al., 2014*). This may explain sequence learning for fear memory (*Wu et al., 2017*) and vicarious trial and error in hippocampus, that is, evaluation of potential paths before decision making (*Johnson and Redish, 2007*; *Singer et al., 2013*). In this case, reverse replay in CA3 contributes to selective forward replay of paths informative for decision making.

In our model, modulation of triggering replay events (or sharp wave ripples) crucially affects the learning with reverse replay. Therefore, other hippocampal sub-regions may also participate in goal-directed path learning. The area CA2 encodes the current position during immobility (*Kay et al., 2016*) and can trigger sharp wave ripples (*Oliva et al., 2016*). Thus, dopaminergic enhancement in CA2 may increase replay events from the current position as in our simulation settings. Dentate gyrus intensively encodes reward sites and triggers sharp wave ripples in CA3 in working memory task (*Sasaki et al., 2018*). It is interesting to examine whether these ripples accompany replay sequences, which remains unclear at present. If this is indeed the case, our results suggest that CA2 and dentate gyrus also play active roles in the goal selection of hippocampal path learning.

Our model also predicts that the release probability of neurotransmitter strongly affects the magnitude and probability of bias toward the reverse direction (*Figure 4*). Therefore, modulation of neurotransmitter release in CA3 can regulate the behavioral impact of hippocampal firing sequences. For example, acetylcholine suppresses neurotransmitter release at recurrent synapses (*Hasselmo, 2006*), which may abolish the directional biases created during movement (see *Figure 4* and *Figure 8—figure supplement 3*). Presynaptic long-term plasticity (*Costa et al., 2015*) may also affect the directional biases at longer timescales.

## Role of theta sequences in hippocampal goal-directed learning

In our simulation, the strength of theta phase-locking of place cell activity did not have significant effects on learning (*Figure 5* and *Figure 5—figure supplement 1*). This result poses a question against the long-standing hypothesis that theta sequences are essential for sequence learning (*Jensen and Lisman, 1996*; *2005*; *Sato and Yamaguchi, 2003*). However, it is possible that theta phase-locking is effective in learning CA3-to-CA1 connections at which STDP is asymmetric and more sensitive to time differences between presynaptic and postsynaptic activities than at CA3 recurrent synapses (*Bi and Poo , 2001*). Furthermore, our simulations in the 2-D space showed that theta sequences can act as readout of weight biases to reward, and hence are useful for planning future trajectories during exploration. This role of theta sequences is consistent with experimental findings (*Huxter et al., 2008*; *Wikenheiser and Redish, 2015*). Short-term facilitation also had only limited effects on the proposed learning mechanism (*Figure 4*); however, it contributed to the generation of theta sequences (*Wang et al., 2015*). Therefore, in our model, short-term facilitation also enhances the readout of spatial information during run.

## Relationships to the previous models

By extending the role of reverse replay in 1-D space, we showed that divergent sequences that isotropically propagate from reward sites helps goal-directed sequence learning and hence efficient

navigation in 2-D space. Similar foraging tasks can be solved by temporal difference (TD) learning model (*Dayan, 1993*; *Sutton, 1988*) and the relevance of TD learning to hippocampal information processing has been proposed (*Foster et al., 2000*; *Stachenfeld et al., 2017*). Thus, how our learning mechanism is related to TD learning should be mathematically investigated in the future.

A conceptual model has been proposed for goal-directed sequence learning with symmetric STDP based on the gradient field of connection strength (*Pfeiffer, 2018*). The conceptual model concluded that goal-directed sequence learning would not occur if sequences homogenously propagate through the entire space. However, our model shows that this is not the case for STP-modulated symmetric STDP and proposes a network mechanism to implement the conceptual model based on reverse replay.

## Some limitations of the present model

While the present model could demonstrate goal-directed path learning, the model has yet to be improved to learn context-dependent switching of behavior such as navigation on an alternating T-maze. In our simulation, animals learn only reward delivered at the same location. However, in alternating T-maze tasks, the animal has to remember recent experiences to change its choices based on the stored memory. Furthermore, the experiment in *Pfeiffer and Foster (2013)* also contains working-memory-based switching between predictable and unpredictable reward searches. Thus, the consistency between our simulations and experiments is still limited. A straightforward extension of our model is to maintain multiple charts representing the alternating paths (*Samsonovich and McNaughton, 1997*) and switch them according to the stored short-term memory or certain context information. Each chart will be selectively reinforced by reverse replay along one of the alternating paths. This switching of CA3 activity may be supported by the dentate gyrus (*Sasaki et al., 2018*). How to protect the previous memories from overwriting with novel reward experiences is another important issue that is left unsolved by the present model. If the animal can memorize all four candidate reward sites simultaneously, the animal can efficiently search reward even when the reward position is changed in every trial. One solution for this issue is triggering remote replay (*Gupta et al., 2010*; *Karlsson and Frank, 2009*) at the previous reward sites. Thus, we predict that bias of starting points of remote replay affects persistence of multiple memories.

While 2-D hippocampal place cells are omnidirectional, the majority of 1-D place cells are unidirectional (*Buzsáki, 2005*). We did not take into account this property of 1-D place cells in this study. Because we only simulated unidirectional movements in the W-maze, the network may describe an ensemble of place cells for one direction and another neuron ensemble is necessary for the opposite direction. In this case, place cells for the path B→D1 and those for the opposite path D1→B are different. Thus, the directional bias learned in our network may not implicate direct paths from D1 to D2 for the animal. To learn unexperienced paths, continuous replay events of multiple unidirectional place-cell ensembles is necessary, which has been experimentally observed (*Davidson et al., 2009*; *Gupta et al., 2010*; *Wu and Foster, 2014*). Neural network models also demonstrated that Hebbian plasticity can connect multiple unidirectional place cells at the junction points (*Brunel and Trullier, 1998*; *Buzsáki, 2005*; *Káli and Dayan, 2000*). Relating to this, replay of long paths and complex spatial structures is often accompanied by concatenated sharp wave ripples (*Davidson et al., 2009*). Interestingly, each sharp wave ripple corresponds to a segment in the spatial structure (*Wu and Foster, 2014*). The effect of this segmentation to our learning mechanism is not obvious. Elucidating the underlying mechanism will reveal how the hippocampal circuit segment and concatenate sequential experiences.

## Possible implications of our results in other cognitive tasks

Reverse replay has not been found in the neocortical circuits. For instance, firing sequences in the rodent prefrontal cortex are reactivated only in the forward directions (*Euston et al., 2007*). To the best of our knowledge, neocortical synapses obey asymmetric STDP (*Froemke et al., 2006*). This seems to be consistent with the selective occurrence of forward sequences because, as suggested by our model, the sensory-evoked forward firing sequences should only strengthen forward synaptic pathways under asymmetric STDP. However, if dopamine turns asymmetric STDP into symmetric STDP, which is actually the case in hippocampal area CA1 (*Brzosko et al., 2015*; *Zhang et al., 2009*), forward firing sequences will reinforce the propagation of reverse sequences. Whether

reverse sequences exist in the neocortex and, if not, what functional roles replay events play in the neocortical circuits are still open questions.

Whether the present neural mechanism to combine experienced paths into a novel path accounts for cognitive functions other than memory is an intriguing question. For instance, does this mechanism explain the transitivity rule of inference by neural networks? The transitive rule is one of the fundamental rules in logical thinking and says, 'if A implies B and B implies C, then A implies C.' This flow of logic has some similarity to that of joint forward-replay sequences, which says, 'if visiting A leads to visiting B and visiting B leads to visiting C, then visiting A leads to visiting C.' Logic thinking is more complex and should be more rigorous than spatial navigation, and little is known about its neural mechanisms. The proposed neural mechanism of path learning may give a cue for exploring the neural substrate for logic operations by the brain.

In sum, our model proposes a biologically plausible mechanism for goal-directed path learning through reverse sequences. In the dynamic programming-based path finding methods such as Dijkstra's algorithm for finding the shortest path (*Dijkstra, 1959*) and Viterbi algorithm for finding the most likely state sequences in a hidden Markov model (*Bishop, 2010*), a globally optimal path is determined by backtrack from the goal to the start after an exhaustive local search of forward paths. Our model enables similar path finding mechanism through reverse replay. Such a mechanism has been suggested in machine learning literature (*Foster and Knierim, 2012*), but whether and how neural dynamics achieves it remained unknown. In addition, our model predicts the roles of neuromodulators that modify plasticity rules and activity levels in sequence learning. These predictions are testable by physiological experiments.

## Materials and methods

### One-dimensional recurrent network model with rate neurons (*Figure 1*)

We simulated the network of 500 neurons. Firing rate of neuron $i$ was determined as

$$r_i = \mathrm{f}_{\mathrm{rate}}\big(I_i^{\mathrm{exc}} - I^{\mathrm{inh}} + I_i^{\mathrm{ext}}\big) \tag{1}$$

The function $\mathrm{f}_{\mathrm{rate}}(I)$ was threshold linear function

$$\mathrm{f}_{\mathrm{rate}}(I) = \max\{0, \rho(I - \epsilon)\} \tag{2}$$

where $\rho = 0.0025$ and $\epsilon = 0.5$. Excitatory synaptic current $I_i^{\mathrm{exc}}$ and Inhibitory feedback $I^{\mathrm{inh}}$ followed

$$\dot{I}_i^{\mathrm{exc}} = -\frac{I_i^{\mathrm{exc}}}{\tau^{\mathrm{exc}}} + \sum_j w_{ij} r_j D_j F_j \tag{3}$$

$$\dot{I}^{\mathrm{inh}} = -\frac{I_i^{\mathrm{inh}}}{\tau^{\mathrm{inh}}} + w^{\mathrm{inh}} \sum_j r_j D_j F_j \tag{4}$$

where $\tau^{\mathrm{exc}} = \tau^{\mathrm{inh}} = 10\,\mathrm{ms}$, and $w^{\mathrm{inh}} = 1$. Variables for short-term synaptic plasticity $D_j$ and $F_j$ obeyed the following equations (*Wang et al., 2015*):

$$\dot{D}_j = \frac{1 - D_j}{\tau_{\mathrm{STD}}} - r_j D_j F_j \tag{5}$$

$$\dot{F}_j = \frac{U - F_j}{\tau_{\mathrm{STF}}} + U(1 - F_j) r_j \tag{6}$$

Parameters were $\tau_{\mathrm{STD}} = 500\,\mathrm{ms}$, $\tau_{\mathrm{STF}} = 200\,\mathrm{ms}$, and $U = 0.6$. External input $I_i^{\mathrm{ext}}$ was usually zero and changed to 5 for 10 ms when the cell was stimulated. We stimulated neurons $0 \leq i \leq 10$ at the beginning of simulations, and neurons $245 \leq i \leq 255$ at 3 s after the beginning. We determined initial values of excitatory weights $w_{ij}$ as

$$w_{ij} = w_{\max} \exp\left(-\frac{|i-j|}{d}\right) \tag{7}$$

where $w_{\max} = 27$ and $d = 5$. We set self-connections $w_{ii}$ to zero throughout the simulations.

The weights were modified according to the rate-based Hebbian synaptic plasticity as

$$\dot{w}_{ij} = \Delta_{ij} \tag{8}$$

$$\tau_{\mathrm{w}}\dot{\Delta}_{ij} = -\Delta_{ij} + \eta r_i r_j D_j F_j \tag{9}$$

where $\eta = 20$ and $\tau_{\mathrm{w}} = 1000$ ms. When we simulated Hebbian synaptic plasticity without the modulation by short-term plasticity, we removed $D_j F_j$ from this equation and changed the value of $\eta$ to 4.

In the simulation with accumulation of ADP (*Figure 1—figure supplement 1*), we calculated smoothed postsynaptic activity $p_i$ by solving

$$\tau_{\mathrm{ADP}}\dot{p}_i = -p_i + r_i \tag{10}$$

where $\tau_{\mathrm{ADP}} = 80$ms, and changed Hebbian plasticity (9) to

$$\tau_{\mathrm{w}}\dot{\Delta}_{ij} = -\Delta_{ij} + \eta p_i r_j \tag{11}$$

The value of $\eta$ was also changed to 4.

## One-dimensional recurrent network model with spiking neurons (*Figure 2*)

We used Izhikevich model (*Izhikevich, 2003b*) for the simulation of spiking neurons:

$$\dot{v}_i = 0.04v_i^2 + 5v_i + 140 - u_i + I_i^{\mathrm{syn}} - I^{\mathrm{inh}} + I_i^{\mathrm{ext}} \tag{12}$$

$$\dot{u}_i = a(bv_i - u_i) \tag{13}$$

If the membrane voltage $v_i \geq 30$ mV, the neuron emits a spike and the two variables were reset as $v_i \leftarrow c$ and $u_i \leftarrow u_i + d$. Parameter values were $a = 0.02$, $b = 0.2$, $c = -65$ and $d = 8$. Excitatory synaptic current was

$$I_i^{\mathrm{syn}} = g_i^{\mathrm{AMPA}}(0 - v_i) + \mathrm{f}_{\mathrm{NMDA}}(v_i)g_i^{\mathrm{NMDA}}(0 - v_i), \tag{14}$$

and the synaptic conductance followed

$$\dot{g}_i^{\mathrm{AMPA}} = -\frac{g_i^{\mathrm{AMPA}}}{\tau_{\mathrm{AMPA}}} + \sum_{j,k} w_{ij}^{\mathrm{AMPA}} D_j F_j \delta(t - t_{jk}^f - t_{\mathrm{delay}}), \tag{15}$$

$$\dot{g}_i^{\mathrm{NMDA}} = -\frac{g_i^{\mathrm{NMDA}}}{\tau_{\mathrm{NMDA}}} + \sum_{j,k} w_{ij}^{\mathrm{NMDA}} D_j F_j \delta(t - t_{jk}^f - t_{\mathrm{delay}}), \tag{16}$$

where $t_{jk}^{\mathrm{f}}$ is the timing of the $k$-th spike of neuron $j$ and parameter values were $\tau_{\mathrm{AMPA}} = 5$ ms, $\tau_{\mathrm{NMDA}} = 150$ ms and $t_{\mathrm{delay}} = 2$ ms. The voltage dependence of NMDA current (*Izhikevich et al., 2004*) was

$$\mathrm{f}_{\mathrm{NMDA}}(V) = \frac{\left(\frac{V+80}{60}\right)^2}{1 + \left(\frac{V+80}{60}\right)^2} \tag{17}$$

Inhibitory feedback $I_i^{\mathrm{inh}}$ was calculated as

$$\dot{I}^{\text{inh}} = -\frac{I_i^{\text{inh}}}{\tau^{\text{inh}}} + w^{\text{inh}} \sum_{j,k} D_j F_j \delta(t - t_{jk}^f - t_{\text{delay}}) \tag{18}$$

where $w^{\text{inh}} = 1$ and $\tau^{\text{inh}} = 10$ ms. Short-term synaptic plasticity obeyed the following dynamics:

$$\dot{D}_j = \frac{1 - D_j}{\tau_{\text{STD}}} - D_j F_j \delta\left(t - t_j^f\right) \tag{19}$$

$$\dot{F}_j = \frac{U - F_j}{\tau_{\text{STF}}} + U(1 - F_j)\delta\left(t - t_j^f\right) \tag{20}$$

Parameter values were $\tau_{\text{STD}} = 500$ ms, $\tau_{\text{STF}} = 200$ ms and $U = 0.6$. External input $I_i^{\text{ext}}$ was the same as that in the rate neuron model.

We determined initial values of synaptic weights $w_{ij}^{\text{AMPA}}$ as

$$w_{ij}^{\text{AMPA}} = w_{\text{max}} \exp(-\frac{|i - j|}{d}) \tag{21}$$

where $w_{\text{max}}$ was 0.3 and $d = 5$. We set self-connections $w_{ii}^{\text{AMPA}}$ to zero throughout the simulations. The weights of NMDA current $w_{ij}^{\text{NMDA}}$ were determined as $w_{ij}^{\text{NMDA}} = 0.2 w_{ij}^{\text{AMPA}}$ and fixed at these values throughout the simulations.

The weights of AMPA current were modified by STDP as

$$\dot{w}_{ij}^{\text{AMPA}} = \Delta_{ij}^{\text{AMPA}} \tag{22}$$

$$\tau_w \dot{\Delta}_{ij}^{\text{AMPA}} = -\Delta_{ij}^{\text{AMPA}} + \eta \sum_{k,l} f_{\text{STDP}}(t_{ik}^f, t_{jl}^f) D_j F_j \delta(t - t_{jk}^f) \tag{23}$$

We simulated two different STDP types by changing the function $f_{\text{STDP}}(t_{\text{post}}, t_{\text{pre}})$ as follows. Asymmetric STDP:

$$f_{\text{STDP}}(t_{\text{post}}, t_{\text{pre}}) = \begin{cases} A_+ \exp(-\frac{t_{\text{post}} - t_{\text{pre}}}{\tau_+}) & \text{if } t_{\text{post}} \geq t_{\text{pre}} \\ -A_- \exp(-\frac{t_{\text{pre}} - t_{\text{post}}}{\tau_-}) & \text{if } t_{\text{post}} < t_{\text{pre}} \end{cases} \tag{24}$$

Symmetric STDP:

$$f_{\text{STDP}}(t_{\text{post}}, t_{\text{pre}}) = A_+ \exp(-\frac{|t_{\text{post}} - t_{\text{pre}}|}{\tau_+}) - A_- \exp(-\frac{|t_{\text{post}} - t_{pre}|}{\tau_-}) \tag{25}$$

Parameter values were $\eta = 0.05$, $\tau_w = 1000$ ms, $A_+ = 1$, $A_- = 0.5$, $\tau_+ = 20$ ms and $\tau_- = 40$ ms for all STDP types. We took into account contributions of all spike pairs were in the simulations. When we simulated STDP without the modulation by short-term plasticity, we removed $D_j F_j$ from the above equation and changed the value of $\eta$ to 0.01.

In the simulation with short time constants (*Figure 2—figure supplement 1A*), we changed the values of parameters as $\tau_{\text{AMPA}} = 2.5$ ms, $\tau^{\text{inh}} = 5$ ms, $w_{\text{max}} = 0.35$ and $w_{ij}^{\text{NMDA}} = 0$.

## Evaluation of weight changes with Poisson spike trains (*Figures 3* and *4*)

In each simulation, we sampled spike trains of 21 neurons (neuron #1 - #21) 100 times for given values of the number of spikes per neuron $N_{\text{spike}}$, mean inter-spike interval (ISI) $t_{\text{ISI}}$, and firing propagation speed $t_{\text{speed}}$. We set first spikes of neuron #$n$ to $t_{n,1}^f = (n - 1)t_{\text{speed}}$. Following a first spike, Poisson spike train for each neuron was simulated by sampling ISI ($\Delta t_{n,k}^f = t_{n,k}^f - t_{n,k-1}^f$) from the following exponential distribution:

$$P\left(\Delta t_{n,k}^f\right) = \frac{1}{t_{\text{ISI}}} \exp(-\frac{\Delta t_{n,k}^f}{t_{\text{ISI}}}), (k = 2, 3, ..., N_{\text{spike}}) \tag{26}$$

We induced an absolute refractory period by resampling ISI if it was shorter than 1 ms. After we generated spike trains, we simulated neurotransmitter release by solving *Equations (19) and (20)* for each neuron. In *Figure 3*, parameter values of short-term plasticity were $\tau_{\mathrm{STD}} = 150\,\mathrm{ms}$, $\tau_{\mathrm{STF}} = 40\,\mathrm{ms}$ and $U = 0.37$. In *Figure 4*, we sampled the values of $U$, $\tau_{\mathrm{STD}}$ and $\tau_{\mathrm{STF}}$ from [0.1, 0.6], [50, 500 ms] and [10ms, 300 ms], respectively.

We calculated changes in the weight from the neuron in the center ($j = 11$) to the neuron $i$ as

$$\Delta_{ij} = \sum_{k,l} \mathrm{f}_{\mathrm{STDP}}(t_{ik}^{\mathrm{f}}, t_{jl}^{\mathrm{f}}) D_j F_j \delta(t - t_{jk}^{\mathrm{f}}) \tag{27}$$

In all-to-all STDP, we calculated the above summation over all spike pairs. In nearest-neighbor STDP (*Izhikevich and Desai, 2003a*), we considered only pairs of a presynaptic spike and the nearest postsynaptic spikes before and after the presynaptic spike. For symmetric STDP, $\mathrm{f}_{\mathrm{STDP}}(t_{\mathrm{post}}, t_{\mathrm{pre}})$ was Gaussian (*Mishra et al., 2016*)

$$\mathrm{f}_{\mathrm{STDP}}(t_{\mathrm{post}}, t_{\mathrm{pre}}) = A \exp(-\frac{1}{2}(\frac{t_{\mathrm{post}} - t_{\mathrm{pre}}}{\tau})^2) \tag{28}$$

where $A = 1$ and $\tau = 70\,\mathrm{ms}$. For asymmtric STDP, $\mathrm{f}_{\mathrm{STDP}}(t_{\mathrm{post}}, t_{\mathrm{pre}})$ was the same as the *Equation (24)* except that parameter values were changed as $A_+ = 0.777$, $A_- = 0.273$, $\tau_+ = 16.8\,\mathrm{ms}$ and $\tau_- = 33.7\,\mathrm{ms}$ (*Bi and Poo , 2001*). We calculated weight biases for each spike train as $\sum_{i=1}^{10} \Delta_{ij} - \sum_{i=12}^{21} \Delta_{ij}$.

## Evaluation of weight changes with theta-modulated Poisson spike trains during run (*Figure 5*)

In each simulation, we sampled one-second-long spike trains of 81 neurons (neuron #1 - #81) 100 times. Calculation of firing probabilities and sampling of spikes were performed every 1 ms time bin. We assumed a constant speed of the rat, and expressed the animal's current position with current time $t$ [s]. We defined theta phase at time $t$ as $\theta(t) = 2\pi \times 8t + c$, where $c$ is a random offset. Place field of each neuron was given by normalized Gaussian function

$$PF_i(t) = \frac{1}{\sqrt{2\pi}\sigma_{\mathrm{PF}}} \exp(-\frac{1}{2}(\frac{t - \mu_i}{\sigma_{\mathrm{PF}}})^2) \tag{29}$$

The place-field center of neuron $i$ was $\mu_i = 2 \times \frac{i-1}{80} - 0.5$ which spanned from -0.5 to 1.5, and $\sigma_{\mathrm{PF}} = 0.2$. We simulated theta phase-locking with von-Mises distribution function (*Bishop, 2010*), which is periodic version of Gaussian function

$$PL_i(t) = \frac{1}{2\pi \mathrm{I}_0(\beta)} \exp(\beta \cos(\theta(t) - p_i(t))) \tag{30}$$

where $\mathrm{I}_0(\beta)$ is the zeroth-order Bessel modified function of the first kind. The mean firing phase of neuron $i$ at time $t$ changed through time as $p_i(t) = \pi(\mu_i - t)$. Using these functions, we determined the firing rate of neuron $i$ at time $t$ as $\alpha PF_i(t) PL_i(t)$ [kHz]. In this setting, $\alpha$ and $\beta$ were the tuning parameters that control the peak firing rates and phase selectivity, respectively. The range of sampling was $0.01 \leq \alpha \leq 0.15$ and $0.1 \leq \beta \leq 10$, and the peak firing rates were calculated by smoothing spike trains of the central neuron (#41) with Gaussian kernel of the standard deviation 50 ms. Changes and biases of weights from the central neuron ($j=41$) were calculated in the same way with the previous section. In the evaluation of the biases summed over 10 overlapping place cells, we sampled 10 spike trains for the central neuron (#41), and summed weight biases calculated from these samples.

## Simulation of learning forward synaptic pathways through reverse replay (*Figure 6*)

We used the same network model as in Figure 1 and additionally implemented normalization of synaptic weights for homeostasis. We calculated the sum of incoming weights on each neuron ($\sum_j w_{ij}$) at each time step and normalized the weights as $w_{ij} \leftarrow \frac{\sum_j w_{ij}^{\mathrm{init}}}{\sum_j w_{ij}} w_{ij}$, where $\{w_{ij}^{\mathrm{init}}\}$ denotes initial synaptic weights. We changed the time constant of long-term plasticity ($\tau_{\mathrm{w}}$) to 5000 ms in the condition 1,

and 500 ms in the conditions 2 and 3. In the condition 3, we also changed the values of parameters for short-term plasticity and initial synaptic weights as $\tau_{\mathrm{STD}} = 200$ ms, $U = 0.3$ and $w_{\max} = 30$. Neurons were stimulated every 1 s. Weight biases were calculated as $\sum_{i<j} w_{ij} - \sum_{i>j} w_{ij}$ for two neurons ($j = 100,\ 400$).

## Simulations of goal-directed sequence learning on a W-maze (*Figures 7* and *8*)

We simulated an animal moving on a two-dimensional space spanned by $x$ and $y$ ($0 \leq x \leq 50,\ 0 \leq y \leq 50$). Coordinates of the six corners (A, B, C1, C2, D1, D2) of W-maze were $\mathbf{z}_A = (25,\ 15)$, $\mathbf{z}_B = (25, 35)$, $\mathbf{z}_{C1} = (45,\ 35)$, $\mathbf{z}_{D1} = (45,\ 15)$, $\mathbf{z}_{C2} = (5, 35)$ and $\mathbf{z}_{D2} = (5,\ 15)$. In each set of trials with time length $T = 15$ s, we determined the position of the animal $\mathbf{z}_{\mathrm{pos}}$ at time $t' = t \bmod T$ as

$$
\mathbf{z}_{\mathrm{pos}} = \begin{cases}
\mathbf{z}_A & (0\,\mathrm{s} \leq t' < 2\,\mathrm{s}) \\
(t' - 2)(\mathbf{z}_B - \mathbf{z}_A) + \mathbf{z}_A & (2\,\mathrm{s} \leq t' < 4\,\mathrm{s}) \\
(t' - 3)(\mathbf{z}_X - \mathbf{z}_B) + \mathbf{z}_B & (4\,\mathrm{s} \leq t' < 6\,\mathrm{s}) \\
(t' - 4)(\mathbf{z}_Y - \mathbf{z}_X) + \mathbf{z}_X & (6\,\mathrm{s} \leq t' < 8\,\mathrm{s}) \\
\mathbf{z}_Y & (8\,\mathrm{s} \leq t' < 15\,\mathrm{s})
\end{cases}
\tag{31}
$$

We set $\mathbf{z}_X = \mathbf{z}_{C1}$ and $\mathbf{z}_Y = \mathbf{z}_{D1}$ in the (2n+1)-th trials and $\mathbf{z}_X = \mathbf{z}_{C2}$ and $\mathbf{z}_Y = \mathbf{z}_{D2}$ in the (2n)-th trials.

The neural network consisted of 2500 place cells that were arranged on a 50 x 50 two-dimensional square lattice. The place field centers of neurons in the $i$-th column (x-axis) and the $j$-th row (y-axis) were denoted as $\mathbf{z}_{i,j} = (i,j)$. Each place cell received excitatory connections from eight surrounding neurons. The connection weight from a neuron at $(i - k, j - l)$ to a neuron at $(i,j)$ were denoted as $w_{i,j}^{k,l}$, where the possible combinations of $(k,l)$ were given as $S = \{(1,1),\ (1,0),\ (1,-1),(0,1),(0,-1),(-1,1),(-1,0),(-1,-1)\}$. Initial connection weights were uniformly random and normalized such that the sum of eight connections obeys $\sum_{(k,l) \in S} w_{i,j}^{k,l} = 0.5$.

We simulated activities of place cells $r_{i,j}$ as

$$
r_{i,j} = \mathrm{f}_{\mathrm{rate}}\left(I_{i,j}^{\mathrm{E}}\right)
\tag{32}
$$

$$
\dot{I}_{i,j}^{\mathrm{E}} = -\frac{I_{i,j}^{\mathrm{E}}}{\tau} + \sum_{(k,l) \in S} w_{i,j}^{k,l} r_{i-k,j-l} D_{i-k,j-l} F_{i-k,j-l} - I^{\mathrm{inh}} - I^{\mathrm{theta}} + I_{i,j}^{\mathrm{place}} + I_{i,j}^{\mathrm{noise}}
\tag{33}
$$

Time constant $\tau$ was 10 ms. The function $\mathrm{f}_{\mathrm{rate}}(I)$ was a threshold linear function

$$
\mathrm{f}_{\mathrm{rate}}(I) = \max\{0, \rho(I - \epsilon)\}
\tag{34}
$$

where $\rho = 1$ and $\epsilon = 0.002$. Inhibitory feedback $I^{\mathrm{inh}}$ followed

$$
\dot{I}^{\mathrm{inh}} = -\frac{I^{\mathrm{inh}}}{\tau^{\mathrm{inh}}} + w^{\mathrm{inh}} \sum_{i,j} r_{i,j} D_{i,j} F_{i,j}
\tag{35}
$$

where $\tau^{\mathrm{inh}} = 10$ ms and $w^{\mathrm{inh}} = 0.0005$. Variables for short-term synaptic plasticity $D_{i,j}$ and $F_{i,j}$ obeyed

$$
\dot{D}_{i,j} = \frac{1 - D_{i,j}}{\tau_{\mathrm{STD}}} - r_{i,j} D_{i,j} F_{i,j}
\tag{36}
$$

$$
\dot{F}_{i,j} = \frac{U - F_{i,j}}{\tau_{\mathrm{STF}}} + U\left(1 - F_{i,j}\right) r_{i,j}
\tag{37}
$$

with parameter values $\tau_{\mathrm{STD}} = 300$ ms, $\tau_{\mathrm{STF}} = 200$ ms, and $U = 0.4$. We induced theta oscillation by

$$
I^{\mathrm{theta}} = \frac{B}{2}(\sin(\frac{2\pi t}{t_{\mathrm{theta}}}) + 1)
\tag{38}
$$

where $B = 0.005\,\text{kHz}$ and $t_{\text{theta}} = \frac{1000}{7}\,\text{ms}$. $I_{i,j}^{\text{noise}}$ was independent Gaussian noise with the standard deviation 0.0005 kHz. We determined place-dependent inputs for each neuron $I_{i,j}^{\text{place}}$ from the place field center of each neuron $\mathbf{z}_{i,j}$ and the current position of the animal $\mathbf{z}_{\text{pos}}$:

$$I_{i,j}^{\text{place}} = C \exp(-\frac{1}{2d^2}(\mathbf{z}_{\text{pos}} - \mathbf{z}_{i,j})^{\text{T}}(\mathbf{z}_{\text{pos}} - \mathbf{z}_{i,j})) \tag{39}$$

where $d = 2$. The parameter $C$ was set as 0.005 kHz when the animal was moving and 0.001 kHz when the animal was stopping at D2 (the position of reward). When the animal was stopping at other positions, $C$ was set at zero but occasionally changed to 0.001 kHz for a short interval of 200 ms. The occurrence of this brief activation followed Poisson process at 0.1 Hz, but it always occurred one second after the onset of each trial to trigger prospective firing sequences.

We implemented the Hebbian synaptic plasticity as

$$\dot{w}_{i,j}^{k,l} = \Delta_{i,j}^{k,l} \tag{40}$$

$$\tau_{\text{w}} \dot{\Delta}_{i,j}^{k,l} = -\Delta_{i,j}^{k,l} + \eta r_{i,j} r_{i-k,j-l} D_{i-k,j-l} F_{i-k,j-l} \tag{41}$$

where $\eta = 1$ and $\tau_{\text{w}} = 30\,\text{s}$. If the sum of synaptic weights on each neuron ($\sum_{(k,l) \in S} w_{i,j}^{k,l}$) was greater than unity, we renormalized synaptic weights by dividing them by the sum. When we simulated Hebbian synaptic plasticity without modulations by short-term plasticity, $D_{i-k,j-l} F_{i-k,j-l}$ was removed from the above equation and the $\eta$ value was changed to 0.1.

We calculated 'connection vector' of each neuron $\mathbf{u}_{i,j}$ by the following weighted sum of unit vectors $\mathbf{v}_{k,l} = \left(\frac{k}{\sqrt{k^2+l^2}}, \frac{l}{\sqrt{k^2+l^2}}\right)$:

$$\mathbf{u}_{i,j} = \sum_{(k,l) \in S} w_{I+k,j+l}^{k,l} \mathbf{v}_{k,l} \tag{42}$$

## Simulations of goal-directed sequence learning in a 2-D space (*Figures 9* and *10*)

We used a similar two-dimensional space and a similar 50 x 50 neural network model to those used in the simulation of the W-maze. We made some minor changes: (1) We normalized initial connection weights as $\sum_{(k,l) \in S} w_{i,j}^{k,l} = 1$. (2) In order to create finite-length firing sequences, we added an external inhibitory input $I^{\text{extinh}}$ to the *Equation (33)*. When the animal was immobile, we kept excitatory input $I_{i,j}^{\text{place}}$ nonzero ($C = 0.001\,\text{kHz}$), and triggered sequences by disinhibiting the network ($I^{\text{extinh}} = 0$) every 1 s and terminated sequences ($I^{\text{extinh}} = 0.1$) at 800 ms after each trigger.

In the simulation of divergent sequences (*Figure 9*), parameters of Hebbian plasticity were changed as $\eta = 0.5$ and $\tau_{\text{w}} = 10\,\text{s}$. We triggered firing sequences at the point $(x, y) = (25, 25)$ for 30 s.

In the simulation of foraging task (*Figure 10*), parameters of Hebbian plasticity were changed as $\eta = 0.1$ and $\tau_{\text{w}} = 10\,\text{s}$. Four candidate reward sites were positioned at $(x, y) = (15, 15), (15, 35), (35, 15), (35, 35)$. The reward position was randomly determined in each simulation. The animal could explore the space defined by $5 \leq x \leq 45$, $5 \leq y \leq 45$. We randomly determined a starting position in this area such that the initial distance to the reward position was longer than 10, and the animal began to explore after 3-s-long immobility. When the animal reached within distance 3 from the reward, the animal was set immobile for 15 s, and then reset to the starting position of the next trial. We terminate a trial when the animal did not reach reward within 300 s, and regarded the exploration time of these trials as 300 s. We excluded these trials from the analysis of angular displacements.

We calculated the activity vector in the 2-D coordinate system as

$$\mathbf{a} = \frac{\sum_{i,j} r_{i,j} \mathbf{z}_{i,j}}{\sum_{i,j} r_{i,j}} - \mathbf{z}_{\text{pos}} \tag{43}$$

when $\sum_{i,j} r_{i,j} > 0$. We changed the animal's position $\mathbf{z}_{\text{pos}}$ through the velocity vector $\mathbf{v}$ as

$$\dot{\mathbf{z}}_{\text{pos}} = \gamma_{\text{v}} \mathbf{v} \tag{44}$$

and rotated the velocity vector towards the direction of the activity vector:

$$\dot{\mathbf{v}} = \gamma_{\text{a}} \frac{\mathbf{a}}{||\mathbf{a}||} + \gamma_{\text{noise}} \mathbf{a}_{\text{noise}} \tag{45}$$

where $\mathbf{v}$ was normalized to unity at each time and $\mathbf{a}_{\text{noise}}$ is a two-dimensional independent normal Gaussian noise. The speed $\gamma_{\text{v}}$ was 0.01 or 0 during exploration and immobility, respectively. Other values of parameters were $\gamma_{\text{a}} = 0.01$ and $\gamma_{\text{noise}} = 0.05$ throughout the simulation. We calculated angular displacements from cosine similarity between $\mathbf{a}$ and the reference vector $\rho$ every 20 ms. Before and during exploration, $\rho$ was a vector from $\mathbf{z}_{\text{pos}}$ to the reward position. At the reward position, $\rho$ was a vector from the current $\mathbf{z}_{\text{pos}}$ to the mean of $\mathbf{z}_{\text{pos}}$ within 3 seconds before reaching the reward position. We excluded the periods in which the maximum firing rate in the network was below 0.01 kHz or the length of $\mathbf{a}$ was below 1. Due to the small size of the network and the 2-D space, angular displacements were not uniform even in the control simulations. To compensate this background biases, we calculated the mean angular displacements of the control simulations and subtracted those baseline values from the mean angular displacements obtained in each condition.

We applied paired sample t-test after checking normality of the data by Shapiro Wilk test ($p > 0.05$).

## Code availability

Simulations and visualization were written in C++ and Python 3. The codes are available at https://github.com/TatsuyaHaga/reversereplaymodel_codes (*Haga, 2018*; copy archived at https://github.com/elifesciences-publications/reversereplaymodel_codes).

## Acknowledgements

We are grateful to Masami Tatsuno, Kaoru Inokuchi, Takeshi Yagi, and members in Fukai lab for fruitful discussion.

## Additional information

### Funding

| Funder | Grant reference number | Author |
| --- | --- | --- |
| Ministry of Education, Culture, Sports, Science, and Technology | 15H04265 | Tomoki Fukai |
| Core Research for Evolutional Science and Technology | JPMJCR13W1 | Tomoki Fukai |
| Ministry of Education, Culture, Sports, Science, and Technology | 16H01289 | Tomoki Fukai |
| Ministry of Education, Culture, Sports, Science, and Technology | 17H06036 | Tomoki Fukai |

The funders had no role in study design, data collection and interpretation, or the decision to submit the work for publication.

### Author contributions

Tatsuya Haga, Conceptualization, Software, Validation, Investigation, Visualization, Writing—original draft; Tomoki Fukai, Conceptualization, Supervision, Funding acquisition, Writing—review and editing

## Author ORCIDs
Tatsuya Haga (iD) http://orcid.org/0000-0003-3145-709X
Tomoki Fukai (iD) http://orcid.org/0000-0001-6977-5638

## Decision letter and Author response
Decision letter https://doi.org/10.7554/eLife.34171.027
Author response https://doi.org/10.7554/eLife.34171.028

# Additional files

## Supplementary files
• Transparent reporting form
DOI: https://doi.org/10.7554/eLife.34171.025

## Data availability
Our study is based on computer simulations, and we are sharing codes in Github.

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
