## [Decision Letter]

Thank you for sending your article entitled "Recurrent network model for learning goal-directed sequences through reverse replay" for peer review at *eLife*. Your article is being evaluated by three peer reviewers, one of whom is a member of our Board of Reviewing Editors, and the evaluation is being overseen by Michael Frank as the Senior Editor.

The reviewers felt that the topic of the paper, to obtain reverse replay through modified STDP rules, was interesting and led to some potentially significant predictions of network behavior. However, there were numerous concerns with the study, particularly relating to clarification of figures and results, and consistency with the literature.

Given the list of essential revisions, which could conceivably involve extensive new work, the editors and reviewers invite you to respond within the next two weeks with an action plan and timetable for the completion of the additional work. We plan to share your responses with the reviewers and then issue a binding recommendation.

1). The authors must clear up several points of confusion in data presentation. a) Figure 6 is confusing in many respects and can be interpreted in differentways from the authors. b) The section on theta modulation and sequences is unclear and must beclarified.

2) The authors must address concerns about apparent inconsistency with the experimental literature. These include the time-course of excitatory input, the presence of forward replays in many studies, the conjunctive coding of space and direction of motion by place cells on linear tracks, and whether replay should correlate with previous experience or future navigation. There are also concerns about whether the learning rules are appropriate for hippocampus, and whether the firing patterns in the model look like in-vivo place cell patterns.

3) The authors should make some more testable predictions, for example, the effect of NMDA knockdown on reverse replay.

These concerns and other suggestions from the reviewers are attached to help the authors.

Reviewer #1:

In this study, the authors show that one can obtain reverse replay in 1 and 2-D networks. This relies on symmetric STDP in combination with various forms of STP or postsynaptic after depolarization. The authors also map their learning rules to a T-maze context using a 2-D network and state that the network learns to do goal-directed path learning through reverse sequences. They examine how network connections organize following such learning. In principle this study is interesting as an implementation of a plasticity-driven approach to the emergence of forward and reverse replay, and provides a way to link it to goal planning.

The initial logic of the paper builds up nicely from Figure 1 through 5. One can see how to obtain reverse replay, there is evidence that this is reasonably robust, and one can see conditions where the replay will erase itself due to plasticity.

Figure 6 is a key figure, applying the learning rule to a 2-D network upon which the authors place a T maze. This is a key figure and unfortunately is very confusing.

1) The authors talk about early and late trials. I am going to assume that only trial 9 and 10 are late trials, but the authors must clarify this point.

2) There is a listing of positions A, B, C1, C2, D1, D2 in Panel A mapping to positions 0, 1, 2, 3, 4, 5 in panels C and D. It is strange to have to jump around with the naming within a single figure.

3) Worse, the mapping is different in odd trials and even trails because position 3 from panel C can either be mapped to D1 (panel A) or to B (panel A). Similarly, position B of panel A could be either position 1 or position 3 of Panels C and D. This makes no sense.

4) The authors make several statements about the reverse replay sequencesthat are hard to identify in the figure. They should individually highlightspecific reverse sequences that they want to talk about.

5) I do not see any case in the later trials where sequences travel to D2 except possibly in the first couple of spontaneous responses in trial 9 and one spontaneous trial in 10. Each of these occurs before the training run. Instead almost all the cases, e.g., in Trial 10, start from D2 and go to B. This is a perfectly reasonable reverse replay but is not presented as such in the text.

6) I do not see any case where backward sequences start from D1 and go to D2, unless the authors are conflating position 3 with D1, rather than B. If so then it is the 3 cases (2 in Trial 9 and 1 in Trial 10, which occur before the training run) which fit the bill. If so, those 3 cases look like forward sequences to me.

7) The text says that some of the reverse replay sequences from D1 propagate into D2 instead of the stem arm. I do not see any instances of this, except again if the authors have confused the identity of position 3.

8) The supplementary video looks interesting but lacks annotation to give clarity. Its value is considerably diminished as a result. Nevertheless, my impression on watching it is that my interpretation of Figure 6 is correct, and that the authors have confused position B and position D1.

9) The key paragraph three in subsection “Goal-directed path learning through reverse replay**”** is very anecdotal. For every statement of a certain kind of replay, the authors need to first, point to examples, and second, give statistics for how often such replays occur in a series of randomized runs.

It may well be that I am quite misunderstanding this figure, in which case the authors should explain it more clearly. Otherwise I think the figure and movie do not support the text.

Reviewer #2:

The manuscript "Recurrent network model for learning goal-directed sequences through reverse replay" proposes an intriguing mechanism for reverse replay of the sequential activation of place cells: combining spike-timing dependent plasticity with synaptic depression, the proposal envisions a wave packet of activity traveling through the network of neurons, such that neurons at the tail end of the packet still fire, but due to synaptic depression, no longer synaptically impinge on the neurons at the front of the packet. As a consequence, Hebbian plasticity strengthens front to back connections, hence enabling reverse replay upon reactivation. In a 2D model, the authors present a nice application of how such a network can produce sequences of place cell activations towards a goal on a path that the animal has never experienced.

The most critical assumption is not, in my opinion, the "rapid modulation of STDP" by synaptic depression, but rather the persistence of neural activity behind the immediate wave-front. Because of profound synaptic depletion, the fact is that there is no reverberant activity that would support the packet and cause neurons to continue to be active.

The trick appears to lie in a time constant of *τ^exc^*=10 ms with which the excitatory input is convolved (Equation 3) Or, in the spiking network, an NMDA time constant of 150 ms (Equation 16), wherein the peak conductance for NMDA is slaved to the AMPA conductance.

The value of 10 ms for the time constant is, at the very least, debatable. Going back to classic papers (Koch, Rapp, Segev, 1996) or Treves (1993), the real excitatory synaptic time constant (as opposed to the membrane time constant of 10-20 ms) is extraordinarily short and on the order of 2 ms. With that kind of time constant, though, I believe the entire mechanism might collapse.

The second critique I would levy is that the model inhabits an intermediate realm: it is neither minimalist, nor veridically detailed.

In particular:

i) Equation 5-6 cover both facilitation and depression. As far as I can tell, facilitation is not at all necessary for the mechanism. Why is facilitation then included?

ii) Equation 12-13 describe the Izhikevich, 2003 model with the parameter set for the regular spiking cell (though the fact that it is the parameter set for RS is not explicitly mentioned). The only possible advantage of the Izhikevich model over an integrate-and-fire model might lie in the adaptation of the firing. But is this important in the model?

iii) Going from 1D to 2D in subsection “Goal-directed path learning through reverse replay**”**, the idea of theta modulation and theta sequences is sprung upon the reader, but it never became clear to me whether Equation 36 (for the theta-modulated current input) is really necessary or not.

The third critique reflects the color scheme. Throughout, inactivity is represented by black, which makes the figures hard to read in a printout or even on the screen. Please choose another color scheme that results in a white (or light-colored) background. Figure 6 is confusing, as it mixes letters on the W-shaped track (which, for some reason, is called a T-shaped track), but then the panels use numeric labels; the mapping is only explained in the last sentence of the caption. On some panels of Supplementary Figure 2, the tick-labels on the y-axes cannot be read.

Reviewer #3:

The paper describes a modified version of spike-timing dependent long-term plasticity (STDP) modulated by short-term synaptic plasticity (STP). The main advantage of this modulation is to be able to obtain an effectively asymmetric STDP rule starting from a symmetric one (symmetric STDP has been recently observed in hippocampus). The authors show for instance how an imposed sequential activation of neurons can modify synapses in a network, so that a network can spontaneously "replay" the sequence in the opposite order, as observed in place cells.

I think that the results nicely characterize the properties of the hypothesized plasticity rule and show a potential application in neural networks.

I have two major concerns:

1) The connection with dynamics of hippocampal place cell activity strongly focuses on reverse replays, but there might be other aspects to consider more carefully:

i) Forward replays: The modeling presented in the first part of the paper, where a reverse replay is elicited by the stimulation of place cell coding for the middle of the track, should be probably compared to the results of Davidson et al., 2009. In that paper, the rodents frequently stop away from the two ends of the track; this more closely resembles the scenario depicted in the manuscript. The results of the paper indicated that forward replays were as (or more) likely to occur compared to backward replays (similar to Diba and Buzsaki, 2007), and the propagation speed of forward and backward replays was comparable. How can one reconcile the results reported in the manuscript with these observations?

ii) Directionality of firing in linear tracks: It is generally observed that place cells code conjunctively for spatial location and direction of motion in 1D. This aspect of place field firing has not been discussed and it's not entirely clear to me how to interpret the authors’ results in light of this experimental observation.

iii) Replays in 2D: Pfeiffer and Foster, 2013 reported replays that generally did not correlate with the previous experience of the animal, rather there was a correlation with the immediate future navigational behavior of the rodent. In those experiments, the goal location was moved from session to session, and those sequences were observed within the first few trials. In a follow-up commentary (Pfeiffer, 2017), it is argued that "reverse replay does not facilitate learning in a familiar environment". It would be useful to see those results and claims more carefully discussed in the manuscript.

iv) Theta sequences: During running, fast sequential activity (forward direction) of place cells within individual cycles of the theta oscillation are observed. I guess that before replays, these sequences could help in building up the asymmetric connections that later generates reverse replays. But what happens to theta sequences after the synapses are modified? Would they be less likely to occur? In general, I found the few references to theta sequences in the manuscript to be confusing.

2) The plasticity rule is loosely based on previous work in visual cortical synapses (Froemke et al., 2006). As such there is no evidence that this rule quantitatively captures the dynamics of synapses in neo-cortex or hippocampus. It would be helpful to test the plasticity rules with firing patterns more closely resembling the activity of in vivo place cells (e.g., fields of ~1s durations, peak firing ~10Hz, phase precessing) and realistic learning rates.

[Editors' note: the authors’ plan for revisions was approved and the authors made a formal revised submission.]

---

## [Author Response]

The reviewers felt that the topic of the paper, to obtain reverse replay through modified STDP rules, was interesting and led to some potentially significant predictions of network behavior. However, there were numerous concerns with the study, particularly relating to clarification of figures and results, and consistency with the literature.1) The authors must clear up several points of confusion in data presentation.a) Figure 6 is confusing in many respects and can be interpreted in differentways from the authors.

Following the reviewers’ suggestions, we have clarified the definitions of types of firing sequences and added quantitative evaluations in Figure 7 (Figure 6 in the previous manuscript). In short, reverse replay refer to sequences propagating backward along the spatial paths that the animal has traveled and joint replay is sequences through unexperienced paths combining two (or more) of experienced paths. Forward replay represents sequences that start from the starting point (A) and propagate along once-traveled paths in the forward direction. We specifically call sequences from A to D2 (reward site) as “goal-directed sequence”. We have confirmed our conclusion by statistics from simulations of 10 model rats (5 rats visited the arms in the reversed visiting order) (Figure 7F-H). These points are explained in paragraph four of subsection “Goal-directed path learning through reverse replay” in the revised manuscript.

We have clarified the relationship between the linearized plots and the three arms of the Y-maze by showing how we linearized the track in Figure 7C (we took a visualization method from Wu and Foster, 2014). We have also added explanations on the structure of the track, the number of trials, the schedule of rat’s movements, reward ON/OFF, and important activity patterns to the movie. In figures including heatmap (ex. Figures 1, 2 and 7), we have changed a color code in which white indicates zero.

b) The section on theta modulation and sequences is unclear and must beclarified.

We have added references for phase precession and theta sequence (e.g. O’Keefe and Recce, 1993; Dragoi and Buzsaki, 2006; Foster and Wilson, 2007), and have performed additional simulations to discuss the effect of phase precession and theta sequence in our model (Figure 5). We have evaluated the weight biases generated by spike trains mimicking place-cell activity with phase precession. The simulation results suggest that a bias to the reverse direction is reliably induced when the peak firing rate >20 Hz (Figure 5B). However, since this value is higher than the average peak firing rate of place cells, we have additionally shown two biologically plausible cases in which significant biases can appear. First, the summation of the bias effects induced by multiple presynaptic place cells with overlapping place fields can result in a significant overall bias in realistic settings of the mean firing rate (Figure 5C). Second, because the firing rates of place cells are log-normally distributed (Mizuseki and Buszaki, 2013), some place cells show higher firing rates and produce large weight changes with large biases. However, we have found that phase precession hardly affects the weight biases (Figure 5B and 5C). All these points are explained in paragraph two of subsection “Bias effects induced by spike trains during run”.

Furthermore, simulations in 2-D space have indicated the role of theta sequences for reading out the learned paths towards the goal (Figure 10), as suggested in experiments (Johnson and Redish, 2007; Wikenheiser and Redish, 2015). These points are explained in paragraph three of subsection “Unbiased sequence propagation enhances goal-directed behavior in a 2-D space”.

2) The authors must address concerns about apparent inconsistency with the experimental literature. These include the time-course of excitatory input, the presence of forward replays in many studies, the conjunctive coding of space and direction of motion by place cells on linear tracks, and whether replay should correlate with previous experience or future navigation. There are also concerns about whether the learning rules are appropriate for hippocampus, and whether the firing patterns in the model look like in-vivo place cell patterns.

As for the time-course of excitatory input, we have included simulation data to show that the bias to the reverse direction occurs even without NMDA current and with shorter AMPA and inhibitory time constants. Furthermore, we have evaluated the weight biases generated by realistic place-cell activity with phase precession, as mentioned above. We have added discussion on the directionality of place cells and the related experimental results (Davidson et al., 2009, Pfeiffer and Foster, 2013 and Pfeiffer, 2017) in paragraph two of subsection “Some limitations of the present model”.

Especially, to connect our model to Pfeiffer and Foster, 2013 and Pfeiffer, 2017, we have additionally simulated learning in a 2D open-field using a 2D neural network (see Figure 9). In a 2D open-field, the directions of propagation of firing sequences from a reward site are often isotropic in our model, and are not limited to the (reverse) direction of the path that the animal has just experienced. This isotropic propagation will create connections converging to the reward site and bias both replays and theta sequences towards the reward site from an arbitrary surrounding point on the field. We suggest that these properties of our model account for experimental observations in open-field exploration tasks (Pfeiffer and Foster, 2013). We have added a new subsection "Unbiased sequence propagation enhances goal-directed behavior in a 2-D space" to explain all these results together with a quantitative evaluation of the directional biases in replay and theta sequences.

3) The authors should make some more testable predictions, for example, the effect of NMDA knockdown on reverse replay.

Experimental evidence suggests that NMDA knockdown impairs the plasticity in the hippocampus and hence degrades the formation of replay (Silva et al., 2016) and the performance of spatial learning task (Morris et al., 1986). In that case, all replay events including reverse replay will disappear and they will become ineffective for learning, whatever the roles of these events are. Therefore, we think that predictions in such a condition may not be meaningful enough. Below, we list other possible predictions of the model, which we have included in the subsection "Testable predictions of the model".

Now, selective control of reverse replay is possible by using the techniques of real-time decoding feedback (Ciliberti and Kloosterman, 2017; Ciliberti, Michon and Kloosterman, 2016). First, our model predicts the consequence of such a control. For instance, if we selectively block reverse replay at a reward site, prospective firing sequences tend to propagate away from the spatial site, and accordingly the animal’s preference to the reward location will be abolished.

Second, as shown in Figure 1A-D, our model predicts that the modulation of STDP by short-term depression is crucial for the preferential strengthening of synaptic pathways reversal to the preceding activity propagation. In addition, based on the plasticity mechanisms shown in Figure 1E and 1F, we expect that this preference should be further enhanced in our model if the time window of symmetric STDP is extended into the acausal temporal domain (t_post_ < t_pre_). We speculate that some neuromodulators (e.g., dopamine or acetylcholine) may cause such a meta-plasticity effect in the STDP of CA3.

Third, modulation of triggering replay events (or sharp-wave ripples) is crucial for goal-directed learning in our model. Recently, it has been revealed that CA2 (Oliva et al., 2016) and dentate gyrus (Sasaki et al., 2018) trigger the significant amount of sharp-wave ripples in the awake state. Therefore, we predict that CA2 and dentate gyrus also play active roles in the goal selection of hippocampal path learning.

These concerns and other suggestions from the reviewers are attached to help the authors.Reviewer #1:In this study, the authors show that one can obtain reverse replay in 1 and2-D networks. This relies on symmetric STDP in combination with various forms of STP or postsynaptic afterdepolarization. The authors also map their learning rules to a T-maze context using a 2-D network and state that the network learns to do goal-directed path learning through reverse sequences. They examine how network connections organize following such learning. In principle this study is interesting as an implementation of a plasticity-driven approach to the emergence offorward and reverse replay, and provides a way to link it to goal planning.The initial logic of the paper builds up nicely from Figure 1 through 5. One can see how to obtain reverse replay, there is evidence that this is reasonably robust, and one can see conditions where the replay will erase itself due to plasticity.

We thank the reviewer for their overall positive comments on our work.

Figure 6 is a key figure, applying the learning rule to a 2-D network upon which the authors place a T maze. This is a key figure and unfortunately is very confusing.

We apologize that our figure was confusing. We have improved the data presentation in the revised manuscript as shown below. Please note that the previous Figure 6 is shown as Figure 7 in the revised manuscript.

1) The authors talk about early and late trials. I am going to assume that only trial 9 and 10 are late trials, but the authors must clarify this point.

Late trials referred to trial 5 to 10 because sequences from D1 to D2 already appear in trial 5. However, we agree that the usage of early and late trials was not clear enough. In the revised manuscript, we showed linearized plots only for trial 5 and 6, which contain all types of sequences we want to discuss. In Figure 7, we have indicated the sequences referred to in the text by colored (red, blue and black) arrows.

2) There is a listing of positions A, B, C1, C2, D1, D2 in Panel A mapping to positions 0, 1, 2, 3, 4, 5 in panels C and D. It is strange to have to jump around with the naming within a single figure.3) Worse, the mapping is _different_ in odd trials and even trails because position 3 from panel C can either be mapped to D1 (panel A) or to B (panel A). Similarly, position B of panel A could be either position 1 or position 3 of Panels C and D. This makes no sense.

These confusions were caused because we mapped two branches of a two-dimensional maze onto a one-dimensional axis. As a consequence, the portions 0→1, 1→3 and 3→5 in the linearized plot refer to the portions A→B, B→D1 and B→D2, respectively, of the actual maze. Thus, the correspondence between the two labeling schemes is not one-to-one, which confused the reviewer. As illustrated in Figure 7C-E, we have improved the data presentation by using the color scheme introduced originally in a related experimental paper (Wu and Foster, 2014).

4) The authors make several statements about the reverse replay sequences that are hard to identify in the figure. They should individually highlight specific reverse sequences that they want to talk about.

We have identified the examples of replay events in all the panels (Figure 7D and 7E) we mention in the text. The following three comments raised by the reviewer have been also clarified in the revised manuscript.

5) I do not see any case in the later trials where sequences travel to D2 except possibly in the first couple of spontaneous responses in trial 9 and one spontaneous trial in 10. Each of these occurs before the training run. Instead almost all the cases, e.g., in Trial 10, start from D2 and go to B. This is a perfectly reasonable reverse replay but is not presented as such in the text.

We apologize to the reviewer for our misleading presentation of replay events. In Figure 7, sequences traveling from A to D2 are the sequences propagating from neuron #1 to #20 (A→B) and successively from neuron #61 to #100 (B→D2) at the initial stages of trials 5 and 6. These sequences are indicated by black arrows. Note that the previous panels for trials 9 and 10 have been removed from the figure and novel panels have been included (Figure 7F-H) to show some quantitative evaluations of the relative frequencies among different firing sequences.

6) I do not see any case where backward sequences start from D1 and go to D2, unless the authors are conflating position 3 with D1, rather than B. If so then it is the 3 cases (2 in Trial 9 and 1 in Trial 10, which occur before the training run) which fit the bill. If so, those 3 cases look like forward sequences to me.

Examples of such sequences are sequences involving neuron #60 to #21 (D1 → B) and neuron #61 to neuron #100 (B → D2). This sequential firing pattern represents one compound sequence. The end point of such a sequence is indicated by a blue arrow in trial 5.

7) The text says that some of the reverse replay sequences from D1 propagate into D2 instead of the stem arm. I do not see any instances of this, except again if the authors have confused the identity of position 3.

Replay events from D2 to D1 is represented by slanted lines starting from neuron #100 and ending at neuron #61 (D2 to B) or by lines starting from neuron #21 and ending at neuron #60 (B to D1). This means that the sequence jumps from neuron #61 to # 21 in the linearized plots. The start point of such a sequence is indicated by a blue arrow in trial 6.

8) The supplementary video looks interesting but lacks annotation to give clarity. Its value is considerably diminished as a result. Nevertheless, my impression on watching it is that my interpretation of Figure 6 is correct, and that the authors have confused position B and position D1.

We have added explanations of each trial to the movie (e.g. the structure of the track, the trial number, the schedule of rat’s movement, reward ON/OFF, explanation of important activity patterns).

9) The key paragraph three in subsection “Goal-directed path learning through reverse replay**”** is very anecdotal. For every statement of a certain kind of replay, the authors need to first, point to examples, and second, give statistics for how often such replays occur in a series of randomized runs.

We have clearly pointed the examples of replay events in a 2-D, and then given the statistics of replay events in the simulations performed with different random seeds and different visiting orders of two arms in paragraph three of subsection “Goal-directed path learning through reverse replay”.

It may well be that I am quite misunderstanding this figure, in which case the authors should explain it more clearly. Otherwise I think the figure and movie do not support the text.

We should agree that the figures and movie were hard to interpret. We hope the improved manuscript resolves all the doubts raised by the reviewer.

Reviewer #2:[…] The most critical assumption is not, in my opinion, the "rapid modulation of STDP" by synaptic depression, but rather the persistence of neural activity behind the immediate wave-front. Because of profound synaptic depletion, the fact is that there is no reverberant activity that would support the packet and cause neurons to continue to be active.The trick appears to lie in a time constant of τ^exc^ =10 ms with which the excitatory input is convolved (Equation 3) Or, in the spiking network, an NMDA time constant of 150 ms (Equation 16), wherein the peak conductance for NMDA is slaved to the AMPA conductance.The value of 10 ms for the time constant is, at the very least, debatable. Going back to classic papers (Koch, Rapp, Segev, 1996) or Treves (1993), the real excitatory synaptic time constant (as opposed to the membrane time constant of 10-20 ms) is extraordinarily short and on the order of 2 ms. With that kind of time constant, though, I believe the entire mechanism might collapse.

We built the model on the basis of bump attractor network models (e.g. Samsonovich and Mcnaughton, 1997; Romani and Tsodyks, 2014). In such models, the generation of localized activity packets (bumps) mainly depends on the distribution of connection weights (e.g., strong localized excitation and weak global inhibition), but the time constant is not important. As for the structure of connection weights, some experiments (e.g., Takahashi et al., 2010; Guzman et al., 2016) suggested that the clusters of strong excitatory connections are overexpressed in CA3, which supports the assumption of strong localized excitation necessary for our model. To show our learning mechanism does not depend on the synaptic time constants, we performed additional simulations with removed NMDA current and shorten AMPA and inhibitory time constants (Figure 2—figure supplement 1).

The second critique I would levy is that the model inhabits an intermediate realm: it is neither minimalist, nor veridically detailed.In particular:i) Equation 5-6 cover both facilitation and depression. As far as I can tell, facilitation is not at all necessary for the mechanism. Why is facilitation then included?

As the reviewer pointed out, facilitation is not necessary for our proposed mechanism. However, we implemented it to evaluate the model’s performance under a realistic situation because strong facilitation is observed in the hippocampus (e.g. Guzman et al., 2016). We could show that our learning mechanism reliably works with a physiologically realistic range of short-term facilitation (Figure 4). We wish to note that in the simulations of the 2D network the facilitation contributes to the generation of theta sequences (Wang et al., 2014).

ii) Equation 12-13 describe the Izhikevich, 2003 model with the parameter set for the regular spiking cell (though the fact that it is the parameter set for RS is not explicitly mentioned). The only possible advantage of the Izhikevich model over an integrate-and-fire model might lie in the adaptation of the firing. But is this important in the model?

In our model, the choice of spiking neuron model is relatively arbitrary because the proposed mechanism does not crucially depend on the specific spiking patterns. Therefore, the use of Izhikevich model is not required. However, the adaptation of firing prevents the persistence of activity packets within the same neuron group, hence making their sequential propagation easier. We briefly mentioned this point in subsection “Potentiation of reverse synaptic transmissions by STDP” in the revised manuscript.

iii) Going from 1D to 2D in subsection “Goal-directed path learning through reverse replay**”**, the idea of theta modulation and theta sequences is sprung upon the reader, but it never became clear to me whether Equation 36 (for the theta-modulated current input) is really necessary or not.

We apologize for the confusion. We have performed additional simulations to discuss the effect of theta sequences on the proposed mechanisms. Simulation results are shown in the subsection "Bias effects induced by spike trains during run", and suggest that theta sequences have almost no effects on learning in our model (Figure 5). However, as suggested in Johnson and Redish, 2007 and Wikenheiser and Redish, 2015, theta sequences are useful for the readout of the learned paths towards the goal. Namely, once synaptic connections are biased towards the position of reward, theta sequences also tend to extend towards the reward, which gives information on the adequate paths to the animal. We have shown this role of theta sequences by simulations in a 2D open-field (Figure 10). We have included references to phase precession and theta sequence (e.g. O’Keefe and Recce, 1993; Dragoi and Buzsaki, 2006; Foster and Wilson, 2007).

The third critique reflects the color scheme. Throughout, inactivity is represented by black, which makes the figures hard to read in a print-out or even on the screen. Please choose another color scheme that results in a white (or light-colored) background. Figure 6 is confusing, as it mixes letters on the W-shaped track (which, for some reason, is called a T-shaped track), but then the panels use numeric labels; the mapping is only explained in the last sentence of the caption. On some panels of Supplementary Figure 2, the tick-labels on the y-axes cannot be read.

We apologize that some figures were not clear enough and Figure 6 (Figure 7 in the revised manuscript) was confusing. Following your suggestions, we have changed the color scheme and labels in the related panels, explained mapping in the linearized plots in a more comprehensive manner. We also corrected the labels in Figure 4—figure supplement 1 (previous Supplementary Figure 2).

Reviewer #3:The paper describes a modified version of spike-timing dependent long-term plasticity (STDP) modulated by short-term synaptic plasticity (STP). The main advantage of this modulation is to be able to obtain an effectively asymmetric STDP rule starting from a symmetric one (symmetric STDP has been recently observed in hippocampus). The authors show for instance how an imposed sequential activation of neurons can modify synapses in a network, so that a network can spontaneously "replay" the sequence in the opposite order, as observed in place cells.I think that the results nicely characterize the properties of the hypothesized plasticity rule and show a potential application in neural networks.

We thank the reviewer for their positive comments on the STDP learning rule proposed in the manuscript.

I have two major concerns:1) The connection with dynamics of hippocampal place cell activity strongly focuses on reverse replays, but there might be other aspects to consider more carefully:i) Forward replays: The modeling presented in the first part of the paper, where a reverse replay is elicited by the stimulation of place cell coding for the middle of the track, should be probably compared to the results of Davidson et al., 2009. In that paper, the rodents frequently stop away from the two ends of the track; this more closely resembles the scenario depicted in the manuscript. The results of the paper indicated that forward replays were as (or more) likely to occur compared to backward replays (similar to Diba and Buzsaki, 2007), and the propagation speed of forward and backward replays was comparable. How can one reconcile the results reported in the manuscript with these observations?

As suggested by the reviewer, our model may explain the weak bias towards forward replays observed in the neural activity recorded in the early stage of learning (Davidson et al., 2009). Once the rat got a reward at the end of the track and reverse replay generated a bias towards the reward site, we expect, from the results of simulations of our model, to observe more forward replays in the middle of the track. Because reverse replay can be observed immediately after the first lap (Foster and Wilson 2006; Wu and Foster, 2014), the bias to the forward direction (i.e., towards the reward) can appear during the very early stage of learning. On the other hand, our evaluation of the bias effect with Poisson spike trains suggests that the bias to the reverse direction is stochastic (not 100% reliable). Therefore, we speculate that the consolidation of strong bias requires repeated experiences. However, we also note that the situation studied in Davidson et al., 2009 is more complex than our simulation setting because they used a very long track to observe long replay sequences across multiple ripples. We briefly mentioned these points in paragraph one of subsection “Testable predictions of the model” and paragraph two of subsection “Some limitations of the present model” in the revised manuscript. We wish to study this interesting problem in the future.

We also wish to leave the problem of propagation speed for our future study because the speed depends on many factors. Propagation speed is directly affected by weight biases in our simple model. However, in the hippocampus, many other factors such as oscillation and dendritic spikes (Jahnke et al., 2015) can affect the propagation speed. Furthermore, some experiment suggests that the propagation speed can be actively controlled without changing sensory inputs (Patalkova et al., 2008). We feel that taking all these factors into account is beyond the scope of this study.

ii) Directionality of firing in linear tracks: It is generally observed that place cells code conjunctively for spatial location and direction of motion in 1D. This aspect of place field firing has not been discussed and it's not entirely clear to me how to interpret the authors’ results in light of this experimental observation.

A straightforward interpretation of our results under unidirectionality is that a 1D track is represented by two subnetworks of unidirectional place cells encoding opposite movement directions at each location. In that case, learning an experienced path (e.g. A to D2 in Figure. 7) is possible in one of the subnetworks through the mechanism we have shown. However, to learn an unexperienced path by combination of multiple paths with different directionality (e.g. D1 to D2 in Figure 7), joint replay through the two subnetworks (B to D1 and B to D2) is necessary. Such replay is experimentally observed (Davidson et al., 2009; Wu and Foster, 2014). The previous studies proposed that Hebbian plasticity supports the generation of joint replay by connecting multiple unidirectional place cells at the junction (Brunel and Trullier, 1998; Káli and Dayan, 2000; Buzsaki, 2005). This issue is related to how the hippocampus connects together and generalizes multiple different episodes and chunks in information streams. We have briefly discussed this issue in paragraph two of subsection “Some limitations of the present model” in the revised manuscript. We will investigate this interesting problem in the future.

iii) Replays in 2D: Pfeiffer and Foster, 2013 reported replays that generally did not correlate with the previous experience of the animal, rather there was a correlation with the immediate future navigational behavior of the rodent. In those experiments, the goal location was moved from session to session, and those sequences were observed within the first few trials. In a follow-up commentary (Pfeiffer, 2017), it is argued that "reverse replay does not facilitate learning in a familiar environment". It would be useful to see those results and claims more carefully discussed in the manuscript.

In our example of 2D neural network, the task was essentially 1D. Therefore, the previous simulations were not really adequate for discussing the implications of our model in navigating a 2D open field. In a 2D open-field, we observed in our model that firing sequences often propagate isotropically from a reward site irrespective of the previous experience. Such an isotropic activity propagation creates 2D connections that converge to the reward site, and hence biases the directions of both forward replays and theta sequences towards the reward site from any position on the field. We think that these properties of our model correspond to the experimental observation in open fields (Pfeiffer and Foster, 2013). In the revised manuscript, we have presented the simulation results of a 2D open-field in Figures 9 and 10 and have discussed the relationship to Pfeiffer and Foster, 2013 and Pfeiffer, 2018 in the subsection "Unbiased sequence propagation enhances goal-directed behavior in a 2-D space". A brief discussion is also found in subsection “Some limitations of the present model” in the Discussion.

iv) Theta sequences: During running, fast sequential activity (forward direction) of place cells within individual cycles of the theta oscillation are observed. I guess that before replays, these sequences could help in building up the asymmetric connections that later generates reverse replays. But what happens to theta sequences after the synapses are modified? Would they be less likely to occur? In general, I found the few references to theta sequences in the manuscript to be confusing.

We have added references for theta sequences (e.g. O’Keefe and Recce 1993; Dragoi and Buzsaki, 2006; Foster and Wilson, 2007). We have performed simulation of place cell activity with phase precession in Figure 5, and the results suggest that the bias is not affected by the existence of phase precession (or theta sequences). The subsection "Bias effect induced by spike trains during run" is devoted to these points. However, we think that theta sequences assist the readout of the learned paths towards the goal (Johnson & Redish 2007; Wikenheiser and Redish, 2015). This is because after synaptic connections are biased, theta sequences tend to spread towards the reward site and gives the animal useful information on the rewarding paths. We have shown this role of theta sequences by simulating navigation in a 2D open-field (Figure 10).

*2) The plasticity rule is loosely based on previous work in visual cortical synapses (Froemke et al., 2006). As such there is no evidence that this rule quantitatively captures the dynamics of synapses in neo-cortex or hippocampus. It would be helpful to test the plasticity rules with firing patterns more closely resembling the activity of* in vivo *place cells (e.g., fields of ~1s durations, peak firing ~10Hz, phase precessing) and realistic learning rates.*

In the revised manuscript, we have evaluated the weight biases generated by place-cell-like activity with phase precession. Our results suggest that biases towards reverse direction are reliably induced when peak firing rate during theta precession firing is greater than 20 Hz. This rate seems to be higher than the average peak firing rate over place cell population. However, summing up bias effects induced in multiple place cells with overlapping place fields resulted in significant bias in the realistic mean firing rate setting (Figure 5B). Furthermore, because the firing rates of place cells are log-normally distributed (Mizuseki and Buszaki, 2013), a small fraction of place cells with high firing rates (~30Hz) will contribute to the generation of weight biases. Our results also suggest that phase precession does not significantly affect the weight bias. This is because the time windows of STDP (70ms) are long enough to eliminate the small influences of different theta-phases in spiking such that only coarse-grained firing rates (place fields) determine the effect of synaptic plasticity (Figure 5—figure supplement 1). These points are discussed in the second paragraph of subsection “Bias effects induced by spike trains during run**”**.